# *Carré du champ* FLOW MATCHING: BETTER QUALITY-GENERALISATION TRADEOFF IN GENERATIVE MODELS

**Jacob Bamberger**[*]
Institute of
Artificial Intelligence
Medical University of Vienna,
University of Oxford

**Iolo Jones**[*]
Institute of
Artificial Intelligence
Medical University of Vienna,
Durham University

**Dennis Duncan**
Institute of
Artificial Intelligence
Medical University of Vienna

**Michael Bronstein**
University of Oxford,
AITHYRA

**Pierre Vandergheynst**
EPFL

**Adam Gosztolai**[†]
Institute of Artificial Intelligence
Medical University of Vienna

## ABSTRACT

Deep generative models often face a fundamental tradeoff: high sample quality can come at the cost of memorisation, where the model reproduces training data rather than generalising across the underlying data geometry. We introduce Carré du champ flow matching (CDC-FM), a generalisation of flow matching (FM), that improves the quality-generalisation tradeoff by regularising the probability path with a geometry-aware noise. Our method replaces the homogeneous, isotropic noise in FM with a spatially varying, anisotropic Gaussian noise whose covariance captures the local geometry of the latent data manifold. We prove that this geometric noise can be optimally estimated from the data and is scalable to large data. Further, we provide an extensive experimental evaluation on diverse datasets (synthetic manifolds, point clouds, single-cell genomics, animal motion capture, and images) as well as various neural network architectures (MLPs, CNNs, and transformers). We demonstrate that CDC-FM consistently offers a better quality-generalisation tradeoff even when used as a latent space generation model. We observe significant improvements over standard FM in data-scarce regimes and in highly non-uniformly sampled datasets, which are often encountered in AI for science applications. Our work provides a mathematical framework for studying the interplay between data geometry, generalisation and memorisation in generative models, as well as a robust and scalable algorithm that can be readily integrated into existing flow matching pipelines.

## 1 INTRODUCTION

Deep generative models aim to sample from an unknown probability density $\nu(x)$, given finitely many training points $\mathcal{D} = \{x^{(i)}\}_{i=1}^N \subset \mathbb{R}^d$. Prominent paradigms include variational autoencoders (Kingma & Welling, 2014), generative adversarial networks (Goodfellow et al., 2014), diffusion processes (Sohl-Dickstein et al., 2015; Ho et al., 2020), and methods based on continuous normalising flows (Chen et al., 2019; Song & Ermon, 2019; Lipman et al., 2023; Albergo et al., 2023) (CNFs). Among these, CNFs have had striking recent success across domains from image generation to molecule design and weather prediction, owing to their ability to generate high-quality samples. However, a trivial and undesirable way of achieving high quality is through reproducing training points or close variants, known as *memorisation*. Thus, in addition to high quality, another desirable property of generative models is their ability to generate novel examples, known as *generalisation*.

Recently, there has been an increasing concern that the observed generative quality has been partly achieved through memorisation (Somepalli et al., 2023; Škrinjar et al., 2025), but possibly on large

---

[*]Equal contribution, ordered alphabetically.
[†]Correspondence: `adam.gosztolai@meduniwien.ac.at`

scales (Graber et al., 2025), undermining novelty, diversity, and data privacy. Recent works have shown that, geometrically, memorisation coincides with the sudden drop or complete vanishing of intrinsic dimensionality of the data manifold (Achilli et al., 2024; Ross et al., 2025; Ventura et al., 2025). In other words, during memorisation, the learned distribution degenerates towards an empirical measure supported on isolated training points, rather than a smooth, finite-dimensional manifold. This observation suggests that a way to address memorisation is by stabilising the intrinsic dimensionality and preserving non-degenerate tangent spaces.

This paper reports an advance on mitigating the quality-generalisation tradeoff in flow matching (FM, Lipman et al. (2023)), a unifying framework of CNFs, that models a deterministic probability path $p_t(x)$ between a source density at $t = 0$ (often Gaussian), and a target density of arbitrary complexity at $t = 1$, and subsumes the probability paths modelled by other generative models, such as diffusion processes and score matching (Lipman et al., 2023). The standard, widely used FM construction induces, near $t = 1$, a *homogeneous and isotropic* Gaussian kernel approximation that concentrates around each training point. In practice, most implementations consider a small-bandwidth limit to maximise accuracy, thereby relying on architecture and training loss for regularisation. Empirically, we find that the quality-generalisation tradeoff defines a frontier for FM: while models stopped early during training typically generalise well but yield subpar sample quality, training longer improves sample quality at the cost of memorisation and, consequently, reduced generalisation. This result remained consistent across datasets and for different neural network architectures (MLP, UNet, Transformer). We further demonstrate that the quality-generalisation tradeoff does not simply depend on the dataset size, but on a balance between local geometry and data sparsity, indicating that memorisation can also occur in large-scale datasets.

To improve the quality-generalisation tradeoff, we introduce Carré du champ flow matching (CDC-FM), which explicitly regularises the FM probability paths through an *anisotropic and inhomogeneous* diffusion term that, for $p_0 = \mathcal{N}(0, \mathbf{I})$, yields the conditional probability path (see Appendix A for the general case for arbitrary initial density $p_0$)

$$p_t(x|x_1) = \mathcal{N}\Big(x; \ t\, x_1, \ \big[(1-t)\,\mathbf{I} + t\,\widehat{\mathbf{\Gamma}}(x_1)^{\frac{1}{2}}\big]^2\Big). \tag{1}$$

The matrix field $\widehat{\mathbf{\Gamma}}$ controls the local Dirichlet (carré du champ) energy, and can be efficiently and robustly estimated from data using diffusion geometry (Jones, 2024a;b; Jones & Lanners, 2026), providing explicit geometric noise regularisation aligned with the data manifold. We demonstrate that across diverse synthetic and real-world datasets and neural network architectures, CDC-FM achieves comparable or better quality than FM, preserving the fine-grained details of the data, while substantially reducing memorisation and increasing generalisation. Our work provides a theoretical framework for the geometry-aware regularisation of flow-based generative models and a practical method that can be readily used in existing FM pipelines.

## 2 BACKGROUND

In this paper, we are interested in modelling a set of data points in $\mathbb{R}^d$ whose density $\nu(x)$ concentrates around an *unknown* lower-dimensional manifold. We begin by revisiting the standard FM formulation (Lipman et al., 2023). We highlight that the FM probability path $p_t(x)$ induces at $t = 1$ a homogeneous, isotropic Gaussian kernel approximation of $\nu$. This will motivate our generalised framework in Section 3, where we incorporate an anisotropic and inhomogeneous kernel approximation to strike a balance between faithfully modelling the data, substantially reducing memorisation, and improving generalisation. To illustrate the distinction between the two frameworks, we use a toy problem of learning to generate a circle from equidistant training points (Fig. 1).

### 2.1 FLOW MATCHING

FM learns a velocity field $u_t(x) : [0, 1] \times \mathbb{R}^d \to \mathbb{R}^d$ (vanishing at the domain boundaries) that generates the probability path $p_t(x)$, satisfying the continuity equation

$$\frac{\partial}{\partial t}p_t(x) = -\nabla \cdot (u_t(x)p_t(x)), \qquad p_0(x) = \mu(x), \, p_1(x) = \nu(x). \tag{2}$$

We consider $\mu(x) = \mathcal{N}(x; 0, \mathbf{I})$ for simplicity, and generalise it to arbitrary $\mu$ in Appendix A. Conceptually, FM takes a bottom-up approach by designing the flow path of particles $X_t := \psi_t(X_0) \sim$

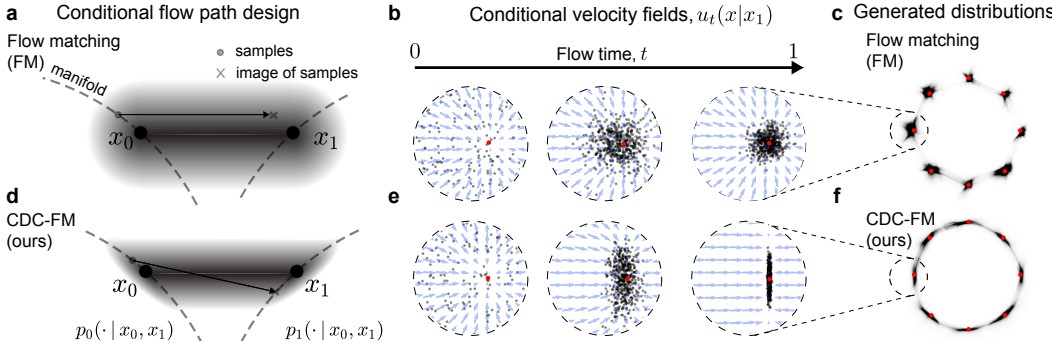

Figure 1: **Carré du champ flow matching. a** FM conditional path design is oblivious to the manifold structure, which can result in off-manifold samples, shown by the black arrows ($\sigma_{\min} > 0$). **b** Conditional velocity fields (blue arrows) in FM transport mass to training points. **c** Generated density by FM trained on eight samples of a unit circle ($\sigma_{\min} = 0$). FM memorises, concentrating likelihood around training points. **d** CDC-FM conditional probability paths are the displacement (optimal transport) interpolants between local covariances and are thus aligned with the geometry. **e** CDC-FM conditional velocity fields flow perpendicular to the manifold. **f** CDC-FM regularises along the manifold, mitigating memorisation and facilitating generalisation.

$p_t$. The velocity field $u_t$ is related to $\psi_t(X_0)$ by the characteristic ODE

$$\frac{d}{dt}\psi_t(x) = u_t(\psi_t(x)), \qquad \psi_0(x) = x. \tag{3}$$

The velocity $u_t$, in turn, induces $p_t(x)$, the probability of finding a particle at $x$ at time $t$, via (2). Specifically, FM takes the final particle position $X_1 \sim \nu$ as an auxiliary variable (a training point) and specifies the conditional flow $\psi_t(X|X_1)$. Although the choice of $\psi_t(X|X_1)$ has implications for the regularity of the density $\nu(x)$ (Albergo et al., 2023), the standard choice is the affine flow

$$\psi_t(X|X_1) = tX_1 + \sigma_t X, \qquad \sigma_t := (1 - t) + t\sigma_{\min}. \tag{4}$$

This choice of flow induces a mapping between Gaussian distributions via the probability path

$$p_t(x|x_1) = \mathcal{N}\left(x \mid tx_1, \sigma_t^2 \mathbf{I}\right), \tag{5}$$

which linearly interpolates the position and convolves with an isotropic Gaussian, and therefore is oblivious to the manifold geometry (Fig. 1a). Given the conditional probability path, the marginal velocity field $u_t(x)$ is learnt via a neural network $\hat{u}_t^\theta(x)$ with weights $\theta$ by minimising the loss

$$\mathcal{L}(\theta) = \mathbb{E}_{t,X_1,X}||\hat{u}_t^\theta(X) - u_t(X|X_1)||^2 = \mathbb{E}_{t,X_0,X_1}||\hat{u}_t^\theta(\psi_t(X_0|X_1)) - \frac{d}{dt}\psi_t(X_0|X_1)||^2, \tag{6}$$

where $t \sim \mathcal{U}[0,1]$, $X_0 \sim \mathcal{N}(0, \mathbf{I})$, $X_1 \sim \nu$, and $X \sim p_t(x|x_1)$. After training, $\hat{u}_t^\theta(x)$ will approximate $u_t(x)$ (Theorem 2, Lipman et al. (2023)), generating the marginal probability path $p_t$ via (2). Note that, in the second equality, we used (3-4) to specify closed-form expressions for the conditional target velocity to lead directly to training points. Our numerical experiments corroborate this, showing that as $t \to 1$ the learnt fields concentrate mass around training points (Fig. 1b).

**Flow matching risks memorisation of training data.** FM's flexibility lives in the transport, i.e., the learnt velocity field $\hat{u}_t^\theta(x)$. Yet, in the limit $t \to 1$, it produces a fixed-bandwidth, isotropic approximation of $\nu$. Indeed, marginalising (5) over the target distribution $\nu(x_1)$ yields the mixture:

$$\nu \simeq p_1(x) = \int_{\mathbb{R}^d} \mathcal{N}\left(x|x_1, \sigma_{\min}^2 \mathbf{I}\right) \nu(x_1)dx_1 \simeq \frac{1}{N}\sum_{i=1}^{N} \mathcal{N}\left(x \Big| x^{(i)}, \sigma_{\min}^2 \mathbf{I}\right). \tag{7}$$

Thus, in the limit $t \to 1$, $\sigma_{\min} \downarrow 0$, the probability path converges to the empirical density. While setting $\sigma_{\min} > 0$ can achieve a fixed-bandwidth regularisation, in practice, it is common to take $\sigma_{\min} = 0$, risking memorisation (Fig. 1c) as shown by simulations of our toy model.

## 3  REGULARISED FLOW MATCHING ON GENERALISED DATA MANIFOLDS

### 3.1  DIFFUSIVE REGULARISATION: CARRÉ DU CHAMP FLOW MATCHING (CDC-FM)

Motivated by the fact that the data density is concentrated around the manifold, we introduce a principled regularisation into FM by replacing the conditional flow path (4) with

$$\psi_t^\Gamma(X|X_1) = tX_1 + \boldsymbol{\Sigma}_t^\Gamma(X_1)^{\frac{1}{2}} X, \qquad \boldsymbol{\Sigma}_t^\Gamma(x) = \left[(1-t)\mathbf{I} + t\widehat{\boldsymbol{\Gamma}}(x)^{1/2}\right]^2, \tag{8}$$

where $\widehat{\boldsymbol{\Gamma}}(x)$ is a local anisotropic covariance around $x$. We have chosen this flow path because it replaces the homogeneous, isotropic covariance in the conditional probability path (5) by

$$p_t(x|x_1) = \mathcal{N}(x \mid tx_1, \boldsymbol{\Sigma}_t^\Gamma(x_1)), \tag{9}$$

the displacement (optimal transport) interpolant between $\mu = \mathcal{N}(0, \mathbf{I})$ and an anisotropic Gaussian centred at $x_1 \sim \nu$ that is geometrically aligned with the data manifold. In Appendix A (Proposition 1), we also derive the conditional probability paths valid for general $\mu, \nu$, illustrated in Fig. 1d.

Note that as an alternative to (8) one could consider the naïve data augmentation that replaces the training points by perturbed samples, $x^{(i)} \mapsto \mathcal{N}(x^{(i)}, \widehat{\boldsymbol{\Gamma}}(x^{(i)}))$, and then uses FM flow paths (5). However, as we prove, this approach generates strictly suboptimal paths (Appendix B, Theorem 1).

Armed with the new conditional path (9), we may directly use the FM loss (6) to obtain the regularised velocity field. Compare Algorithms 1 and 2 in Appendix C for a condensed summary. A simple approximation shows that (8) provides an inductive bias for the learnt velocity field to approximate the data manifold. To see this, we substitute our flow path (8) into the FM loss (6), noting that the target velocity is $\frac{d}{dt}\psi_t(X_0|X_1 = x_1) = x_1 + \left[\widehat{\boldsymbol{\Gamma}}(x_1)^{1/2} - \mathbf{I}\right]X_0$. The random part (second term) can be approximated to leading order as $\left[\widehat{\boldsymbol{\Gamma}}(x_1)^{1/2} - \mathbf{I}\right]X_0 \sim \mathcal{N}(0, \mathbf{I} - 2\widehat{\boldsymbol{\Gamma}}(x_1)^{1/2} + \widehat{\boldsymbol{\Gamma}}(x_1)) \simeq \mathcal{N}(0, \mathbf{I} - \widehat{\boldsymbol{\Gamma}}(x_1))$. This means that if $\widehat{\boldsymbol{\Gamma}}(x_1)$ approximates the projection map onto the local tangent space, the dominant contributions to the velocity are approximately perpendicular (Fig. 1e), minimising tangential flows (Fig. 1c), which are associated with memorisation (Achilli et al., 2024).

To understand the mechanism behind the geometric regularisation induced by the flow path (8), we may first focus on the target distribution $\nu(x_1)$. As before, we marginalise (9) to obtain

$$\nu \simeq p_1(x) = \frac{1}{N}\sum_{i=1}^N \mathcal{N}\left(x \Big| x^{(i)}; \widehat{\boldsymbol{\Gamma}}(x^{(i)})\right). \tag{10}$$

We see that our flow path replaced the FM approximation of the data manifold $\nu$ in (7) with an anisotropic Gaussian mixture. As $\widehat{\boldsymbol{\Gamma}}(x_1)$ approximates the projection map onto the local tangent space, we numerically observe that CDC-FM faithfully learns the data manifold (Fig. 1f) in contrast to FM (Fig. 1c).

To obtain a deeper understanding of the regularisation mechanism, one may study marginal probability paths $p_t$. We prove (see Appendix D, Proposition 2) that our flow path (8) induces a geometry-aware anisotropic diffusion term to the continuity equation (2) to obtain a drift-diffusion process (22). For diffusion processes, one may use the Dirichlet energy (32) to quantify the smoothing introduced by the diffusion term arising from our flow paths. However, the Dirichlet energy is just the CDC field $\widehat{\boldsymbol{\Gamma}}(x^{(i)})$, measuring the local smoothness around a training point $x^{(i)}$, integrated over the data manifold. This justifies the construction of our flow path.

### 3.2  ESTIMATING THE CARRÉ DU CHAMP

To compute $\widehat{\boldsymbol{\Gamma}}$ that optimally captures the local geometry, we follow Jones (2024a;b); Jones & Lanners (2026) and provide a local kernel density estimate using the diffusion maps Laplacian (Coifman & Lafon, 2006; Berry & Harlim, 2016). We compute a variable-bandwidth Gaussian kernel up to the $k$th neighbour of node $i$, $w_\epsilon(x^{(i)}, x^{(j)}) = \exp\left(-\|x^{(i)} - x^{(j)}\|^2/(\epsilon_i \epsilon_j)\right)$, where $\epsilon_i, \epsilon_j$ is the distance to the $k_{bw}$th nearest neighbour of $x^{(i)}, x^{(j)}$, respectively. We use this to obtain a local estimate of the transition probabilities of a Markov process generating the data:

$$(Pf)(x^{(i)}) := \sum_j P_{ij}\, f(x^{(j)}) \ = \ \mathbb{E}_{Y \sim P_i}[f(Y)], \qquad P_{ij} := \frac{w_\epsilon(x^{(i)}, x^{(j)})}{\sum_\ell w_\epsilon(x^{(i)}, x^{(\ell)})}. \tag{11}$$

where $f$ is a well-behaved test function and $P_{ij}$ represents the one-step transition probabilities from sample $x^{(i)}$ to a neighbour $x^{(j)}$. Using the local Markov kernel estimates (11), we compute

$$\widehat{\mathbf{\Gamma}}(x^{(i)}) = \mathbb{E}_{X \sim P_i} \left[ \left( X - m^*(x^{(i)}) \right) \left( X - m^*(x^{(i)}) \right)^T \right], \qquad (12)$$

which is the local covariance of the random variable $X \sim P_i$ (Bakry et al., 2014). We prove in Appendix E (Theorem 2) that (12) is the optimal Gaussian covariance at $x^{(i)}$ given the Markov kernel (11). In practice, we downscale $\widehat{\mathbf{\Gamma}}(x^{(i)})$ to ensure that the added Gaussians (10) add only a small first-order correction to the FM path and do not distort the training distribution (Appendix E). We then take the rank-$d_{cdc}$ approximation of $\widehat{\mathbf{\Gamma}}$, optimising $d_{cdc}$ using grid search. To globally scale the effect of regularisation, we also introduce a hyperparameter $\gamma$ multiplying $\widehat{\mathbf{\Gamma}}$.

**Computational complexity.** Our algorithm is scalable to large datasets. The additional component compared to FM is the computation of $\widehat{\mathbf{\Gamma}}$, which has a complexity of $\mathcal{O}(N \log(N))$ and a memory requirement of $\mathcal{O}(N)$ with respect to the training set size (Appendix F). Further, in our experiments below, we also report the number of function evaluations (NFE) required for the adaptive ODE solver to reach a prespecified tolerance during inference. We find that CDC-FM has comparable or lower inference-time complexity than FM.

## 4 EXPERIMENTS

We now present a series of experiments to quantify the advantages of geometric regularisation over FM, particularly concerning: (i) low *memorisation of training data*; (ii) good *generalisation to test data*; and (iii) high sample *quality*. We quantify (i) by marking a generated sample $y$ from the model as memorised if its nearest-neighbour ratio (Yoon et al., 2023), $M(y) := ||y - x^{(1)}||/||y - x^{(2)}||$, with $x^{(1)}$ and $x^{(2)}$ the first and second nearest training neighbours of $y$, falls below a cutoff. To allow analysis at the level of training points, we compute the percentage of memorised samples per nearest training point and average for a global memorisation measure. To measure (ii), we use the negative log-likelihood (NLL) of a test dataset, which is equivalent to the cross-entropy loss between the data and the model predictions. While both (i) and (ii) indirectly measure quality (iii), we, in addition, use the distance-to-manifold (DtM) when possible. For details on network architectures and hyperparameters, see Appendix G.

### 4.1 IMPROVED REPRESENTATION OF GEOMETRIC DATASETS

Given that $\widehat{\mathbf{\Gamma}}$ approximates tangent spaces, we expected our method to be well-suited for data with strong geometric structure.

**Three-dimensional geometry inference.** Light Detection and Ranging (LiDAR) scans provide point clouds of complex two-dimensional (2D) surfaces embedded in 3D space from limited samples. We consider topographic LiDAR data from Mt. Rainier, WA, previously used in geometric applications of FM (Liu et al., 2024; Kapusniak et al., 2024). We trained FM and CDC-FM on 40-200 uniformly sampled points, with velocity fields parameterised by multilayer perceptrons.

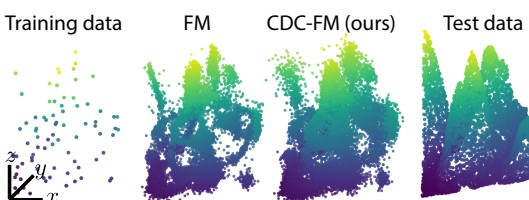

Figure 2: **Visual comparison of FM vs CDC-FM for LiDAR data.**

Early in training, both FM and CDC-FM samples covered the target manifold, achieving low memorisation and good generalisation but poor geometric fidelity (DtM, Table A6). As training progressed, their behaviour diverged. FM had consistently higher quality than CDC-FM, but achieved this by memorising training points, at the expense of generalisation. In contrast, CDC-FM improved quality with substantially better generalisation (Table A7). Qualitatively, FM terrain reconstructions appeared patchy and disconnected, while those of CDC-FM were smoother and coherent (Fig. 2).

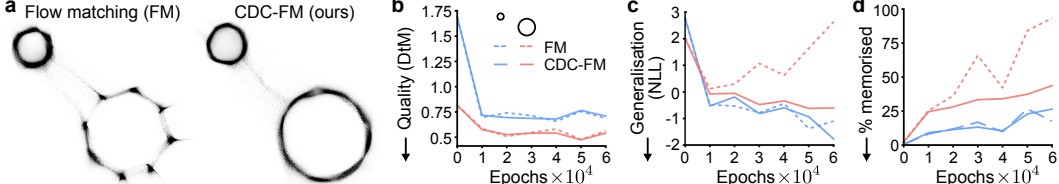

Figure 3: **Early stopping for spatially heterogeneous data. a** Samples from FM and CDC-FM trained on the two-circles dataset at an epoch late in the training (40k), when FM captures the small circle and memorises samples on the larger one. **b** Quality, **c** generalisation, and **d** memorisation against training epoch for the two methods, presented separately for the two circles. Lines represent means over samples.

Table 1: **Comparison of FM and CDC-FM on single-cell data.** Earth mover distance between predicted and held-out snapshots, mean over 5 runs.

| Method | Cite ↓ | Multi ↓ |
|---|---|---|
| I-FM | $48.276 \pm 3.281$ | $57.262 \pm 3.855$ |
| I-CDC-FM | $46.657 \pm 3.412$ | $54.419 \pm 0.629$ |
| OT-FM | $45.393 \pm 0.416$ | $54.814 \pm 5.858$ |
| OT-CDC-FM | $44.410 \pm 0.993$ | $52.043 \pm 1.948$ |

**Inference of single-cell gene expression trajectories.** We evaluated the impact of geometric regularisation on two single-cell gene expression benchmarks (CITE-seq – *'Cite'* and Multiomics – *'Multi'*) from Lance et al. (2022). These datasets comprise temporal snapshots of low-dimensional, spatially complex trajectories in a high-dimensional space, where points define the gene expression state of cells. Interpolation between snapshots is challenging because cell sampling methods are destructive, meaning cells between time points are unpaired.

Following Kapusniak et al. (2024), we used PCA to reduce dimensionality to 100-dimensions, and performed leave-one-out interpolation by assessing the models' ability to reconstruct from a total of four snapshots one of the two intermediate snapshots. We modelled the velocity field using an MLP and trained FM and CDC-FM until the validation loss plateaued. We found that CDC-FM resulted in consistently better reconstruction than FM, both when samples from snapshots were unpaired (I-FM) or paired using optimal transport (Tong et al. (2024), OT-FM, Table 1).

These experiments demonstrate that CDC-FM can improve the quality-generalisation tradeoff in domains with underlying geometric structure.

## 4.2 QUALITY-GENERALISATION TRADEOFF FOR SPATIALLY HETEROGENEOUS DATA

Choosing the number of training epochs trades off sample quality against generalisation and memorisation. We hypothesised that for spatially heterogeneous data, different regions converge at different rates: at any fixed epoch, FM may memorise sparse regions and generalise in dense ones.

**Early stopping for spatially heterogeneous data.** To illustrate this, we trained FM and CDC-FM on two circles with different diameters, each with eight training points. We observed two training phases. In the initial phase ($\lesssim 10^4$ epochs), both methods broadly covered both circles (Fig. A1a) with low quality (Euclidean DtM; Fig. 3b), corroborating our earlier findings. In the second phase ($\gtrsim 10^4$ epochs), sample quality improved rapidly (Fig. 3a,b). For FM, this improvement on the sparser, larger circle came via collapse onto training points (Fig. 3a), reflecting the loss of generalisation (Fig. 3c) and memorisation (Fig. 3d). By contrast, CDC-FM quality increased without over-representation of training points, yielding markedly less memorisation (Fig. 3d), stable generalisation (Fig. 3c) and better inference-time numerical efficiency (Fig. A1b) across both circles.

Overall, this example illustrates that there is no single optimal early-stopping point for FM. At any epoch, FM tends to produce a mix of high-quality yet memorised samples and novel yet low-quality samples. CDC-FM is substantially less sensitive to this heterogeneity, allowing training to proceed until the desired quality is reached without incurring memorisation.

**Limiting spatially localised memorisation in animal motion capture data.** To reinforce our results on the two-circles data, we considered animal motion capture data of the fruit fly, *Drosophila melanogaster* (DeAngelis et al., 2019). Points in this data are 31-frame pose sequences where each snapshot represents the 2D position of the six legs relative to the body centroid (Fig. 4a, inset), yielding a 372-dimensional ($2 \times 6 \times 31$) state space. A 3D UMAP embedding of this data permits

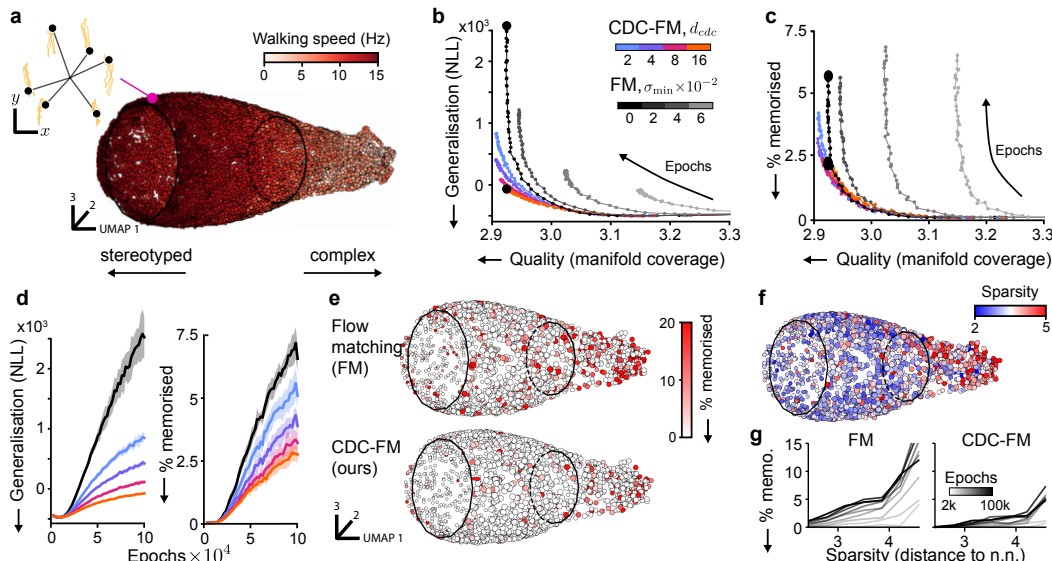

Figure 4: **Quality-generalisation tradeoff for animal motion capture data. a** Inset: an example fruit fly pose sequence. Point cloud, each point representing a 31-frame pose sequence, visualised in 3D UMAP coordinates. Shading indicates walking speed. **b** Generalisation against quality for CDC-FM for different $\widehat{\mathbf{\Gamma}}$ ranks, $d_{cdc}$, and for FM for different $\sigma_{\min}$. Points are averaged over eight seeds. Black circles indicate epochs analysed in e. **c** Same as b, but for memorisation. **d** Generalisation and memorisation against epochs. Shaded error bars show one std from the mean. **e** Percentage of memorised samples nearest to a training point (FM: $\sigma_{\min} = 0$, CDC-FM: $d_{cdc} = 16$). **f** Variation of train data sparsity. **g** Average memorisation against sparsity for different epochs.

the visualisation of a data manifold, which is parametrised by the walking speed on the longitudinal axis and phase on the cyclic coordinate (Fig. 4a). We also performed a sensitivity analysis for the parameters $k_{bw}$ (bandwidth) and $k$ (number of nearest neighbours), finding that only $k$ had measurable influence on the estimation of the CDC field (Fig. A5). We obtained better generalisation and lower memorisation than FM for orders of magnitude changes of $k$. Nevertheless, optimal regularisation can be achieved for $k$ values large enough to provide an accurate kernel density estimate, but small enough to avoid short-cut connections in the manifold (Fig. A5).

Using the pose sequences as training points, we trained transformers, a leading architecture for character motion synthesis (Hu et al., 2023), to approximate the generative velocity field of CDC-FM and FM (with different $\sigma_{\min}$). We measured quality by the faithfulness with which samples covered the manifold, quantified by the mean distance of test points to the nearest samples. We found that FM with $\sigma_{\min} = 0$ traced out a frontier at increasing training epochs, trading off sample quality with generalisation (Fig. 4b) and memorisation (Fig. 4c). We used a memorisation cutoff of $M(y) \simeq 0.6$, confirmed by the movement traces (Fig. A2). Naïvely regularising FM by increasing $\sigma_{\min}$ did not exceed this frontier, achieving either worse generalisation (memorisation) or quality. By contrast, adding CDC regularisation ($\gamma > 0$) surpassed the FM frontier, simultaneously improving sample quality, generalisation and memorisation. While the advantage was strongest around $\gamma = 0.3$ (Fig. 4b,c), other values also lead to improvements (Fig. A3). Increasing $d_{cdc} = rank(\widehat{\mathbf{\Gamma}})$ led to better generalisation with a slight drop in quality, possibly due to noise leakage in off-manifold directions.

Amongst the models in Fig. 4b,c across different epochs, which one should one choose? When plotted against epochs, we see that generalisation in FM monotonically deteriorates (Fig. 4d) and memorisation monotonically increases. This means that FM models require early-stopping strategies to optimise. By contrast, CDC-FM test-set performance plateaus and memorisation remains lower, meaning that early-stopping strategies are less relevant. Our results show that for a given sample quality, especially when high, there is a CDC-FM model with comparable or better generalisation and memorisation than FM, and vice versa.

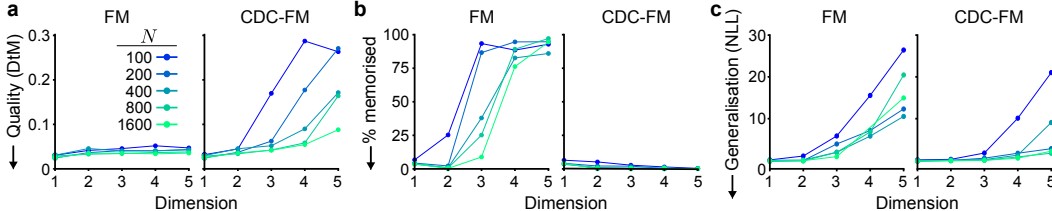

Figure 5: **Synthetic experiment on toroidal manifold.** Effect of data dimension on **a** sample quality, **b** memorisation and **c** generalisation.

The data manifold is not covered uniformly with data points. Rather, fast behaviours are generally more stereotypical and most densely represented (left side of manifold) than slow, complex movements (right side of manifold, Fig. 4a). We expected that this sparsity pattern would lead to varying trade-offs in quality, generalisation and memorisation. In visualising these patterns, we took a late epoch (100k) for both FM and CDC-FM (Fig. 4b,c, black dots), finding that memorisation predominantly occurred in sparse regions (Fig. 4e). Memorisation correlated with the sparsity of the training points (Fig. 4f,g), estimated based on the distance to the closest neighbour, and occurred first at the sparsest points (Fig. 4g), corroborating our earlier findings (Fig. 3). By contrast, CDC-FM showed substantially lower (Fig. 4f,g) and less sparsity-dependent memorisation (Fig. 4g). Overall, these results indicate that geometry-aware regularisation is especially valuable when data varies heterogeneously over the data manifold. In the next section, we examine its limits as the intrinsic dimension and dataset size increase.

## 4.3 LIMITS OF EXPLICIT GEOMETRIC REGULARISATION

Until now, we have demonstrated that CDC-FM achieves better quality-generalisation tradeoff for different architectures (MLP, UNet, Transformers) and diverse domains (LiDAR, single-cell, motion capture). We now test the limits of CDC-FM on two axes, which are known to challenge manifold methods: (i) increasing dimensionality, (ii) scaling to large data.

**Influence of manifold dimension.** To study the influence of dimension, we generate tori $T^d = (S^1)^d \subset \mathbb{R}^{2d}$ varying the dimensions $d$ and the number of training points. As the dimension increases, the effective data density decreases exponentially (curse of dimensionality), and we expected the performance of generative models to decrease. We present the results for FM and CDC-FM trained to 4k epochs, but found that qualitatively similar results hold for different epochs.

As before, we found that FM samples were of consistently high quality, and while CDC-FM could attain a comparable quality, it required an increasingly larger number of training points as the dimension increased (Fig. 5a). Yet, strikingly, this increased quality by FM was a result of memorising almost all of the dataset when the dimension was high enough (Fig. 5b). This shows that memorisation is more likely to occur for higher dimensions for a fixed data size. By contrast, while memorisation in FM and CDC-FM was comparable in dimensions one or two, CDC-FM memorisation decreased with dimension and remained low (Fig. 5b). CDC-FM also attained generally higher generalisation (Fig. 5c). Taken together, these experiments show that, unlike FM, CDC-FM prevents memorisation and facilitates generalisation irrespective of dimension, an effect that remained robust when adding small Gaussian noise to the data (Fig. A4). Yet, as the dimension increases, the desirable sample quality may only be achievable with a sufficient amount of data, likely due to the $k$nn graph construction of the kernel.

**Addressing the curse of dimension through latent diffusion: Celeba-HQ.** We then trained FM and CDC-FM in latent space, and decoded the output back to pixel space. To test whether our findings extend to such a setting, and following the setup of Dao et al. (2023), we evaluate FM and CDC-FM in the latent space of the Stable Diffusion VAE (Rombach et al., 2022). We train both models on a subset of 1000 Celeba-HQ images, and report FID and

Table 2: **FID and NLL per epoch for FM vs. CDC-FM in latent space.** CelebA-HQ subset of size 1000.

| Epoch | FID ↓ | | NLL ↓ | |
|---|---|---|---|---|
| | FM | CDC-FM | FM | CDC-FM |
| 1000 | 15.60 | 12.72 | 6.80 | 7.18 |
| 2000 | 13.51 | 13.42 | 6.78 | 6.83 |
| 3000 | 13.56 | 10.55 | 6.80 | 6.68 |
| 4000 | 13.82 | 11.70 | 6.69 | 6.53 |
| 5000 | 13.10 | 10.85 | 6.68 | 6.48 |

NLL across epochs in Table 2. Despite the dimensionality reduction to size $32 \times 32 \times 4$, we observe that the CDCFM regularisation improves both quality and generalisation after 3k epochs once both model performances stabilise.

**Influence of dataset size.** Finally, we studied the scaling of CDC-FM to large data in image synthesis, where generative models have enjoyed considerable success, although memorisation is still reported, particularly for a low number of training points (Kadkhodaie et al., 2024; Gu et al., 2025).

We trained FM on increasing subsets of CIFAR-10 for $\sigma_{\min} = 0$, as we found that this choice yielded the best performance (Fig. A6), and similarly, CDC-FM, using a UNet network architecture. We measured sample quality by the Fréchet inception distance (FID) on a test set of 10k samples. We found that, for small training set sizes ($<$10k), all training points became abruptly memorised ($M <$ 0.2) as training progressed, consistent with a phase transition as reported in diffusion models (Pham et al., 2024), while the fraction of memorised points steadily increased with larger training sizes (Fig. 6b). By comparison, we found only a few per cent of memorised points with CDC-FM (Fig. 6b). Lower memorisation was accompanied by substantially better generalisation and sample quality at a late epoch after the FM phase transition (4k, Fig. 6c,d, bottom), but only comparable generalisation and quality for epochs before the phase transition (2k, Fig. 6c,d, top). Yet, as the number of training points increased, quality and generalisation in FM and CDC-FM were comparable across epochs, indicating that implicit regularisation, from the architecture and loss function, becomes dominant.

Our results demonstrate that the benefit of geometric noise is highest for low, heterogeneous or geometrically structured data settings.

## 5 RELATED WORK

**Manifold hypothesis and generative modelling.** Our approach is motivated by geometric approaches of generative modelling under the so-called manifold hypothesis, surveyed in Loaiza-Ganem et al. (2024), which presumes that high-dimensional data often have low-dimensional structure. Generative models have also been used to estimate intrinsic geometric quantities, like dimension (Stanczuk et al., 2024). Conversely, related line of work constrains generative models to predefined manifolds (Chen & Lipman, 2024; Huang et al., 2022; Mathieu & Nickel, 2020; De Bortoli, 2022), or to manifolds that are learnt from data (Kapusniak et al., 2024; Peach et al., 2024; Gosztolai et al., 2025). Our use of the carré du champ is based on diffusion geometry (Jones, 2024a), which generalises classical Riemannian geometry to complex geometries that do not meet the strict definition of a manifold. Rather than using the geometry as a constraint, we use it as a form of model regularisation by modifying the probability path with an anisotropic diffusive term aligned to the data geometry.

**The quality-generalisation tradeoff in diffusion models.** Memorisation and generalisation have been studied in the context of diffusion models, where empirical evidence showed a strong dependence on training set size and architecture (Yoon et al., 2023; Gu et al., 2025). From a geometric standpoint, Ross et al. (2025) argues that memorisation occurs when the learnt manifold's dimension is too low due to either overfitting-driven memorisation, when tangent directions are not fully captured, or data-driven memorisation, when the underlying data manifold itself is degenerate. In this regard, Achilli et al. (2024) observed failures to capture tangent space dimensions and Ventura et al. (2025) studied the dynamical regimes of diffusion models from a geometric perspective. On the theoretical side, De Bortoli (2022) studied the convergence of denoising diffusion models under manifold assumptions. While FM and diffusion share strong formal analogies — indeed, FM unifies score-based and diffusion training — there has been much less work on memorisation in FM. Bertrand et al. (2025) shows that the optimal FM vector field memorises and argues that generalisation does not come from the stochasticity of the FM objective. Our results thus complement and extend the existing understanding of memorisation phenomena in generative modelling.

**Geometric regularisation.** Early work applied geometric regularisation to supervised learning by introducing tangent information either via hand-crafted invariances (Simard et al., 1991) or tangents estimated from neural network Jacobians (Rifai et al., 2011). CDC-FM differs from these by applying geometric regularisation in a generative modelling setting: rather than modifying the loss,

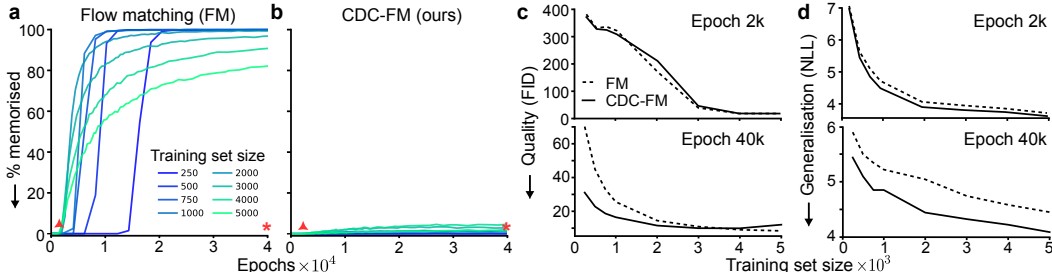

Figure 6: **CIFAR-10 in the low data regime.** Percentage of points memorised for **a** FM and **b** CDC-FM for increasing training epochs. **c** Quality and **d** generalisation against training set at a representative early (2k, orange triangle) and late (40k, orange star) epoch.

## 6 DISCUSSION

We introduced Carré du Champ Flow Matching (CDC-FM), a principled modification of flow matching (FM) that injects geometric regularisation into the conditional probability path. Our generalisation can be understood as adding a spatially-modulated, geometry-aware noise to the deterministic FM generative path to minimise its Dirichlet energy. This noise term encourages transport normal to the data manifold and suppresses tangential collapse onto training points. We show that this noise term can be rigorously justified as the optimal transport path in the space of probabilities and can be optimally estimated from data.

Empirically, we find across synthetic and real data that sample quality and generalisation are in tradeoff. Namely, FM models require a specified number of training epochs to reach a desired sample quality, which comes at a cost of memorisation and decreased generalisation to the test set. CDC-FM could mitigate this tradeoff by achieving higher generalisation for the desired sample quality for diverse datasets, including geometric point clouds (LiDAR) and continuous trajectories (single-cell time courses; motion capture). Notably, CDC-FM reduced localised memorisation that plagues FM under heterogeneous sampling densities, even for moderately large data.

Our model uses a form of the manifold hypothesis based on local heat kernel approximations. This is weaker than the typical Riemannian approximation, as it allows for the intrinsic dimension to vary and is robust to noise. Yet, we found that as the manifold dimension rises, our method needed exponentially more samples to maintain accuracy, due to the need to estimate tangent spaces from local neighbourhoods. Further, for non-geometric data, the benefit of geometric regularisation diminished, on average, as the training set size increased, because the neural network architectures and the loss function already confer inductive biases. Yet, we showed that architecture-induced regularisation can depend heavily on the local complexity and sparsity of the data, while the use of CDC-FM remained robust and resulted in better generalisation and, in some cases, also higher quality in diverse data settings than FM.

Overall, we expect that even in overall high data settings, local memorisation is likely a common occurrence, driven by local sparsity patterns (Škrinjar et al., 2025). In these scenarios, we expect that CDC-FM can provide a robust tool to reduce or eliminate memorisation and improve generalisation. Thus, our approach is not a competitor to FM but a plug-in regulariser that can be scheduled, adapted, or even learned and can be readily used in conjunction with existing latent space pipelines. While we used our geometric framework to generalise FM, being one of the most broadly adopted generative models, future work could explore the effect of our geometric regularisation on other generative models and stochastic regularisation strategies. Our work thus opens a path to geometry-aware flow-based generative models with stronger guarantees, better sample efficiency, and improved robustness to privacy risks.

## 7 ACKNOWLEDGEMENTS

This work is supported by an ERC grant (NEURO-FUSE, Project DOI: 10.3030/101163046). MB and JB are partially supported by the EPSRC Turing AI World-Leading Research Fellowship No. EP/X040062/1 and EPSRC AI Hub No. EP/Y028872/1.

## 8 AUTHOR CONTRIBUTIONS

The work was conceived and performed at the Institute of Artificial Intelligence of the Medical University of Vienna, except for refining results and manuscript writing. JB, IJ, PV and AG conceived the study. JB, IJ and AG developed the mathematical formalism. JB, IJ and DD performed the experiments. JB, IJ, DD and AG wrote the manuscript with comments from PV and MB.

## 9 REPRODUCIBILITY STATEMENT

All details necessary to reproduce the experiments are present in the paper. Code is available at https://github.com/Dynamics-of-Neural-Systems-Lab/cdc-fm.

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

# Appendices

## USE OF LLMs AND OTHER TOOLS

We used Grammarly to facilitate writing and GPT-5, Gemini, and Claude to check for mathematical and notational inconsistencies, as well as facilitate writing code.

## A  ARBITRARY SOURCES AND TARGETS

In the main text, we described our method in the special case of generating flow paths from a Gaussian $\mu = \mathcal{N}(0, \mathbf{I})$ to a complex density $\nu$. Here we show that our framework can be readily extended to model flow paths between two arbitrary densities $\mu, \nu$. In addition to the single-cell dataset in Section 4.1, this setup is relevant for many other applications (Liu, 2022; Albergo et al., 2023). FM naturally lends to this setting as the conditional paths can be conditioned on both source and target samples (Tong et al., 2024). First, to approximate the manifold structure, we compute CDC matrices $\widehat{\boldsymbol{\Gamma}}(x_0), \widehat{\boldsymbol{\Gamma}}(x_1)$, for training points $x_0 \sim \mu, x_1 \sim \nu$, respectively, as described in Section 3.2. We incorporate these local tangent spaces in the conditional probability path by using the displacement interpolant between $\mathcal{N}(x_0, \widehat{\boldsymbol{\Gamma}}(x_0))$ and $\mathcal{N}(x_1, \widehat{\boldsymbol{\Gamma}}(x_1))$. We formalise this in the following proposition.

**Proposition 1.** *Given that $p_0(x|x_0, x_1) = \mathcal{N}(x_0, \widehat{\boldsymbol{\Gamma}}(x_0))$ and $p_1(x|x_0, x_1) = \mathcal{N}(x_1, \widehat{\boldsymbol{\Gamma}}(x_1))$ are Gaussian, the displacement interpolant between them, $p_t(x|x_0, x_1) := [p_0, p_1]_t$, is also Gaussian, with mean and covariance given by*

$$
\begin{aligned}
\mu_t &= (1-t)x_0 + tx_1 \\
\Sigma_t &= \mathbf{A}_t \widehat{\boldsymbol{\Gamma}}(x_0) \mathbf{A}_t^\top
\end{aligned}
\tag{13}
$$

*where $\mathbf{A}_t = (1-t)\,\mathbf{I} + t\mathbf{B}$ and $\mathbf{B} := \widehat{\boldsymbol{\Gamma}}(x_0)^{-\frac{1}{2}} \left( \widehat{\boldsymbol{\Gamma}}(x_0)^{\frac{1}{2}} \widehat{\boldsymbol{\Gamma}}(x_1) \widehat{\boldsymbol{\Gamma}}(x_0)^{\frac{1}{2}} \right)^{\frac{1}{2}} \widehat{\boldsymbol{\Gamma}}(x_0)^{-\frac{1}{2}}$, with the corresponding conditional flow being*

$$
\psi_t(x|x_1, x_0) = (1-t)x_0 + tx_1 + \mathbf{A}_t \left( x - x_0 \right).
\tag{14}
$$

*Proof.* Follows from the optimal transport interpolation of two covariance matrices $\widehat{\boldsymbol{\Gamma}}(x_0), \widehat{\boldsymbol{\Gamma}}(x_1)$ (Bhatia et al., 2019). $\qquad\square$

## B  CARRÉ DU CHAMP FLOW MATCHING GENERATES OPTIMAL TRANSPORT FLOW PATHS

Since we aim to improve the approximation of the underlying density as $\nu(x) \simeq \frac{1}{N} \sum_{i=1}^N \mathcal{N}(x^{(i)}, \widehat{\boldsymbol{\Gamma}}(x^{(i)}))$ using the carré du champ tangent space approximations $\widehat{\boldsymbol{\Gamma}}(x^{(i)})$, a naïve approach might be to randomly augment each training point $x^{(i)}$ by some perturbation from $\mathcal{N}(0, \widehat{\boldsymbol{\Gamma}}(x^{(i)}))$, take the standard conditional OT paths (5), and train against the FM loss (6). However, we show that this approach is equivalent to FM between $\mathcal{N}(0, \mathbf{I})$ and $\mathcal{N}(x^{(i)}, \widehat{\boldsymbol{\Gamma}}(x^{(i)}))$ with non-optimal transport probability paths.

**Theorem 1.** *Training an FM model with source points $X_0 \sim \mathcal{N}(0, \mathbf{I})$ and target points $Z \sim \mathcal{N}(X_1, \widehat{\boldsymbol{\Gamma}}(X_1))$ for $X_1 \sim \nu$ is equivalent, in expectation, to training with the conditional probability path*

$$
\tilde{p}_t(x|x_1) = \mathcal{N}(tx_1, (1-t)^2 \mathbf{I} + t^2 \widehat{\boldsymbol{\Gamma}}(x_1))
\tag{15}
$$

*which differs from the optimal probability path obtained via displacement interpolation*

$$
p_t(x|x_1) := [\mathcal{N}(0, \mathbf{I}), \mathcal{N}(x_1, \widehat{\boldsymbol{\Gamma}}(x_1))]_t = \mathcal{N}\left(tx_1, \left[(1-t)\mathbf{I} + t\widehat{\boldsymbol{\Gamma}}(x_1)^{1/2}\right]^2\right)
\tag{16}
$$

*whenever $\widehat{\boldsymbol{\Gamma}}(x_1) \neq 0$.*

*Proof.* We first show that training with conditional flow paths leading to a naïvely augmented training set leads to the correct marginal probability paths. We start by considering the FM loss with augmented data, which results in the loss

$$\mathcal{L}_{\text{aug}}(\theta) := \mathbb{E}_{t, X_0, X_1, Z} \| \hat{u}_t^\theta ((1-t)X_0 + tZ) - (Z - X_0) \|^2 \tag{17}$$

for $t \sim \mathcal{U}(0,1)$, $X_0 \sim p$, $X_1 \sim \nu$, and $Z \sim \mathcal{N}(x_1, \widehat{\mathbf{\Gamma}}(x_1))$. Differentiating with respect to $\theta$, we get

$$\nabla_\theta \mathcal{L}_{\text{aug}} = \mathbb{E}_{X_1} \nabla_\theta \mathbb{E}_{t, X_0, Z} \| \hat{u}_t^\theta ((1-t)X_0 + tZ) - (Z - X_0) \|^2. \tag{18}$$

The expectation on the right is just the FM loss (6) for an affine flow $\widetilde{\psi}_t(X|Z) = (1-t)X + tZ$ between $\mathcal{N}(0, \mathbf{I})$ and $\delta_Z$. By Theorem 2 in Lipman et al. (2023), we may marginalise over the conditional flows $\widetilde{\psi}_t(\cdot|Z)$

$$\nabla_\theta \mathbb{E}_{t, X_0, Z} \| \hat{u}_t^\theta ((1-t)X_0 + tZ) - (Z - X_0) \|^2 = \nabla_\theta \mathbb{E}_{t, X_0} \| \hat{u}_t^\theta (\widetilde{\psi}_{t, X_1}(X_0)) - \frac{d}{dt} \widetilde{\psi}_{t, X_1}(X_0) \|^2 \tag{19}$$

where $\widetilde{\psi}_{t, X_1}$ denotes the marginal flow between $\mathcal{N}(0, \mathbf{I})$ and $\mathcal{N}(x_1, \widehat{\mathbf{\Gamma}}(x_1))$. This tells us that we can view data augmentation as a distinct FM problem for each training point $X_1$. We may then take the expectation over these points to obtain

$$\nabla_\theta \mathcal{L}_{\text{aug}} = \mathbb{E}_{X_1} \nabla_\theta \mathbb{E}_{t, X_0} \| \hat{u}_t^\theta (\widetilde{\psi}_{t, X_1}(X_0)) - \widetilde{\psi}'_{t, X_1}(X_0) \|^2$$
$$= \nabla_\theta \mathbb{E}_{t, X_0, X_1} \| \hat{u}_t^\theta (\widetilde{\psi}_{t, X_1}(X_0)) - \widetilde{\psi}'_{t, X_1}(X_0) \|^2.$$

If we define conditional flows between $\mathcal{N}(0, \mathbf{I})$ and $\mathcal{N}(x_1, \widehat{\mathbf{\Gamma}}(x_1))$ by $\widetilde{\psi}_t(\cdot|X_1) := \widetilde{\psi}_{t, X_1}$ then

$$\nabla_\theta \mathcal{L}_{\text{aug}} = \nabla_\theta \mathbb{E}_{t, X_0, X_1} \| \hat{u}_t^\theta (\widetilde{\psi}_t(X_0|X_1)) - \widetilde{\psi}'_t(X_0|X_1) \|^2, \tag{20}$$

which shows that training with $\mathcal{L}_{\text{aug}}$ is the same, in expectation, as training with $\mathcal{L}$ (6) using the conditional flows $\widetilde{\psi}_t(\cdot|X_1)$.

Although the above shows that naïve data augmentation is equivalent to an FM problem, we now show that the resulting conditional flow paths are different from the CDC-FM paths, and so are suboptimal. We can derive a closed-form expression for the conditional path $\tilde{p}_t(\cdot|X_1)$ as

$$\tilde{p}_t(\cdot|X_1) = \mathcal{N}\big(tX_1, (1-t)^2\mathbf{I} + t^2\widehat{\mathbf{\Gamma}}(X_1)\big), \tag{21}$$

because, if $X \sim \tilde{p}_t(\cdot|X_1)$, then $X = ta + (1-t)b$ where $a \sim \mathcal{N}(0, \mathbf{I})$ and $b \sim \mathcal{N}(x_1, \widehat{\mathbf{\Gamma}}(x_1))$ are independently sampled, but the optimal probability path between $\mathcal{N}(0, \mathbf{I})$ and $\mathcal{N}(x_1, \widehat{\mathbf{\Gamma}}(x_1))$ is given by the displacement interpolant

$$p_t(\cdot|X_1) := [\mathcal{N}(0, \mathbf{I}), \mathcal{N}(X_1, \widehat{\mathbf{\Gamma}}(X_1))]_t$$
$$= \mathcal{N}\big(tX_1, \big[(1-t)\mathbf{I} + t\widehat{\mathbf{\Gamma}}(X_1)^{1/2}\big]^2\big)$$
$$= \mathcal{N}\big(tX_1, (1-t)^2\mathbf{I} + t^2\widehat{\mathbf{\Gamma}}(X_1) + 2t(1-t)\widehat{\mathbf{\Gamma}}(X_1)^{1/2}\big),$$

which differs from $\tilde{p}_t(\cdot|X_1)$ whenever $\widehat{\mathbf{\Gamma}}(X_1) \neq 0$. Therefore, whereas the probability paths used in CDC-FM are displacement (optimal transport) interpolants, those in FM with data augmentation are not displacement interpolants, except for when $\widehat{\mathbf{\Gamma}}(X_1) = 0$, where they both collapse to FM. □

This means that, unlike the displacement interpolant paths in CDC-FM, data augmentation is not conditionally optimal. In Lipman et al. (2023), the authors show that an FM model trained with deterministic displacement interpolant probability paths requires fewer solver steps to compute a solution.

## C PSEUDO CODE

We present pseudo code for CDC-FM in the case of noise to data and compare it to standard FM in the following algorithms.

| **Algorithm 1** Flow Matching | **Algorithm 2** CDC Flow Matching |
|---|---|
| 1: **Input:** Samples $\mathcal{D} = \{x^{(i)}\}_{i=1}^N \subset \mathbb{R}^d$ from $\nu(x)$ parameterised vector field $v^\theta(x,t)$. | 1: **Input:** Samples $\mathcal{D} = \{x^{(i)}\}_{i=1}^N \subset \mathbb{R}^d$ from $\nu(x)$ together with their local covariance square-root $\widehat{\Gamma}(x^{(i)})^{0.5}$; parameterised vector field $v^\theta(x,t)$. |
| 2: **Output:** Gradient $\nabla_\theta \mathcal{L}(\theta)$ for updating parameters $\theta$ | 2: **Output:** Gradient $\nabla_\theta \mathcal{L}(\theta)$ for updating parameters $\theta$ |
| 3: Sample $x_1 \leftarrow x^{(i)} \sim \nu(x)$ | 3: Sample $x_1 \leftarrow x^{(i)} \sim \nu(x)$ |
| 4: Sample $t \sim \text{Uniform}[0,1]$ | 4: Sample $t \sim \text{Uniform}[0,1]$ |
| 5: Sample $x_0 \sim \mathcal{N}(0, \mathbf{I})$ | 5: Sample $x_0 \sim \mathcal{N}(0, \mathbf{I})$ |
| 6: Define | 6: Define |
| $$x_t \leftarrow tx_1 + ((1-t) + t\sigma_{\min}) x_0$$ | $$x_t \leftarrow tx_1 + \left((1-t)\,\mathbf{I} + t\widehat{\Gamma}(x_1)^{0.5}\right) x_0$$ |
| 7: Define the target conditional vector field at $$v_t(x_t \mid x_1) \leftarrow x_1 - (1 - \sigma_{\min}) x_0$$ | 7: Define the target conditional vector field $$v_t(x_t \mid x_1) \leftarrow x_1 - \left(\mathbf{I} - \widehat{\Gamma}(x_1)^{0.5}\right) x_0$$ |
| 8: Compute the loss $$\mathcal{L}(\theta) = \left\| v^\theta(x_t, t) - v_t(x_t \mid x_1) \right\|^2$$ | 8: Compute the loss $$\mathcal{L}(\theta) = \left\| v^\theta(x_t, t) - v_t(x_t \mid x_1) \right\|^2$$ |
| 9: Compute the gradient $\nabla_\theta \mathcal{L}(\theta)$ | 9: Compute the gradient $\nabla_\theta \mathcal{L}(\theta)$ |
| 10: **return** $\nabla_\theta \mathcal{L}(\theta)$ | 10: **return** $\nabla_\theta \mathcal{L}(\theta)$ |

## D  INTERPRETATION OF CDC-FM AS AN ANISOTROPIC DIFFUSIVE REGULARISATION

Fundamentally, our framework seeks to identify the Markov process, whose density at $t = 1$ matches the data density $\nu$. The CDC-FM flow path (8) introduces a data-driven regularity in this Markov process. In this section, we justify that the CDC-FM flow path is equivalent to adding a space-dependent, anisotropic diffusion term into the marginal probability path (2). This leads to the Fokker–Planck equation

$$\frac{\partial}{\partial t} p_t(x) = (L^* p_t)(x) := -\nabla \cdot (u_t(x)\, p_t(x)) + \frac{1}{2} \sum_{q,r} \partial_q \partial_r (A_t(x)_{qr}\, p_t(x)). \tag{22}$$

Here, $L^*$ is the adjoint of $L$, the infinitesimal generator of the stochastic process, and $A_t$ is the diffusion tensor, which adapts to the local geometry of data.

Note that the mean and covariance in (8) do not depend on $x$, only on the end-point $x_1$. We may therefore prove the following result.

**Proposition 2.** *Let $p_t(x)$ define a Gaussian probability path*

$$p_t(x) = \mathcal{N}(x;\, m_t, \Sigma_t), \qquad m_t \in \mathbb{R}^d,\ \Sigma_t \in \mathbb{S}^d_{++}, \tag{23}$$

*where $m_t, \Sigma_t$ are differentiable and independent of $x$. Then, $p_t(x)$ satisfies the following Fokker-Planck equation*

$$\frac{\partial}{\partial t} p_t(x) = \nabla \cdot (\dot{m}_t\, p_t(x)) + \frac{1}{2} \sum_{q,r} \partial_q \partial_r (\dot{\Sigma}_{t,q,r}\, p_t(x)). \tag{24}$$

*Proof.* Because $p_t(x)$ is Gaussian, we have

$$\log p_t(x) = C - \tfrac{1}{2} \log \det \Sigma_t - \tfrac{1}{2}(x - m_t)^\top \Sigma_t^{-1}(x - m_t). \tag{25}$$

Differentiating with respect to $t$, we have

$$\frac{\partial}{\partial t} \log p_t = \frac{1}{p_t} \frac{\partial}{\partial t} p_t = -\tfrac{1}{2}\text{tr}(\Sigma_t^{-1}\dot{\Sigma}_t) + \dot{m}_t^\top \Sigma_t^{-1}(x - m_t) + \tfrac{1}{2}(x - m_t)^\top \Sigma_t^{-1}\dot{\Sigma}_t \Sigma_t^{-1}(x - m_t). \tag{26}$$

Further, for a Gaussian density $p_t(x)$, we may compute the drift term

$$-\nabla \cdot (\dot{m}_t \, p_t) = \dot{m}^\top \boldsymbol{\Sigma}^{-1} (x - m_t) p_t. \tag{27}$$

Likewise, for the diffusion part

$$\tfrac{1}{2} \sum_{q,r} \partial_q \partial_r (\dot{\Sigma}_{t,qr} \, p_t) = \tfrac{1}{2} \sum_{q,r} \dot{\Sigma} \partial_q \partial_r p_t = \tfrac{p_t}{2} \left[ (x - m_t)^\top \boldsymbol{\Sigma}_t^{-1} \dot{\boldsymbol{\Sigma}}_t \boldsymbol{\Sigma}_t^{-1} (x - m_t) - \mathrm{tr}(\boldsymbol{\Sigma}_t^{-1} \dot{\boldsymbol{\Sigma}}_t) \right]. \tag{28}$$

Adding (27) and (28), we obtain the right-hand side of (26), which is the desired result. $\square$

Using Proposition 2, we see that the conditional probability path satisfies

$$\frac{\partial}{\partial t} p_t(x|x_1) = \nabla \cdot (\dot{m}_t \, p_t(x|x_1)) + \frac{1}{2} \sum_{q,r} \partial_q \partial_r (\dot{\Sigma}_{t,q,r} \, p_t(x|x_1)), \tag{29}$$

where, using our CDC-FM flow path in (8), we have

$$\begin{aligned} \dot{m}_t(x_1) &= x_1 \\ \dot{\boldsymbol{\Sigma}}_t(x_1) &= \left[ (1-t)\mathbf{I} + t\widehat{\boldsymbol{\Gamma}}(x_1)^{1/2} \right] (\widehat{\boldsymbol{\Gamma}}(x_1)^{1/2} - \mathbf{I}). \end{aligned} \tag{30}$$

We may then marginalise by taking expectations over $\nu$ in (29) to obtain (22) with

$$\begin{aligned} u_t(x) &= \mathbb{E}_\nu[X_1 | X_t = x] \\ A_t(x) &= \mathbb{E}_\nu[\dot{\boldsymbol{\Sigma}}_t(X_1) | X_t = x], \end{aligned} \tag{31}$$

where we used the fact that, by Bayes' rule, $\int F(x_1) p_t(x|x_1) p_1(x) dx_1 = p_t(x) \mathbb{E}_\nu[F(X_1)|X_t = x]$ for any integrable function $F$.

**Dirichlet form and carré du champ.** The global smoothness introduced by the diffusive term induced by the flow path is measured by the (weighted) Dirichlet form $\mathcal{E}(f, g)$, which monotonically decreases along the evolution $p_t$. For well-behaved test functions $f, g : \mathbb{R}^d \to \mathbb{R}$ this reads

$$\mathcal{E}(f, g) := \int_{\mathbb{R}^d} \Gamma(f, g)(x) \, dx := \frac{1}{2} \int_{\mathbb{R}^d} (L(fg) - fLg - gLf) \, dx = \int_{\mathbb{R}^d} \nabla f \cdot A_t(x) \nabla g \, dx. \tag{32}$$

In contrast to $\mathcal{E}$, the integrand $\Gamma(f, g)$, known as the carré du champ (CDC), measures the local, point-wise smoothness, and encodes the geometry of the underlying manifold (Jones, 2024a). Note, we recover FM in the deterministic limit, $A_t(x) \equiv 0$ as $t \to 1$. Consequently, $\mathcal{E} = 0$, meaning the associated generator has no intrinsic mechanism to dissipate energy, thus no smoothing.

Taken together, the spatially varying covariances introduce a source of randomness into the sample paths, which can be interpreted as an anisotropic diffusion. The CDC quantifies this diffusion and relates it to the smoothness of the data itself.

# E    OPTIMAL ESTIMATION OF THE CARRÉ DU CHAMP MATRIX

In Section 3.2, we estimate the local geometry at a sample $x$ using the carré du champ matrix $\widehat{\boldsymbol{\Gamma}}(x)$, which we define as the mean-centred covariance of the sample's neighbours. We now prove that this choice is optimal in the sense that it is the best local Gaussian approximation to the data. Specifically, the Gaussian that maximises the expected log-likelihood of the kernel $P_x$ is the one whose mean and covariance match the empirical mean and covariance.

**Theorem 2.** *Let $P_{xy}$ be a transition kernel on $\mathbb{R}^d$. Let $P_x$ denote the probability measure supported on the $k$ nearest neighbours of $x$ obtained by restricting and renormalising $P_{xy}$ (11). Assume $P_x$ has finite second moments. Then the unique maximiser over $m \in \mathbb{R}^d$ and $\Sigma$ over the space of positive definite matrices of the expected log-likelihood of a sample $Y$ from a Gaussian density $\mathcal{N}(m, \boldsymbol{\Sigma})$*

$$(m, \boldsymbol{\Sigma}) \longmapsto \mathbb{E}_{Y \sim P_x}[\log \mathcal{N}(Y; m, \Sigma)]$$

*is given by matching the first two moments of $P_x$:*

$$m^*(x) = \mathbb{E}_{Y \sim P_x}[Y], \qquad \boldsymbol{\Sigma}^*(x) := \widehat{\boldsymbol{\Gamma}}(x) = \mathbb{E}_{P_x}\left[ (Y - m^*)(Y - m^*)^\top \right].$$

*Equivalently, $\mathcal{N}(m^*, \boldsymbol{\Sigma}^*)$ minimises $\mathrm{KL}(P_x \,\|\, \mathcal{N}(m, \boldsymbol{\Sigma}))$.*

*Proof.* For $\Sigma \succ 0$, the log-likelihood of a sample $Y \sim \mathcal{N}(m, \Sigma)$ is

$$\log \mathcal{N}(Y) = -\frac{d}{2}\log(2\pi) - \frac{1}{2}\log \det \Sigma - \frac{1}{2}(Y - m)^\top \Sigma^{-1}(Y - m).$$

Taking the expectation with respect to $Y \sim P_x$ gives

$$\mathbb{E}_{Y \sim P_x}[\log \mathcal{N}(Y; m, \Sigma)] = -\frac{d}{2}\log(2\pi) - \frac{1}{2}\log \det \Sigma - \frac{1}{2}\mathbb{E}_{Y \sim P_x}\left[(Y - m)^\top \Sigma^{-1}(Y - m)\right].$$

We can differentiate with respect to $m$ to get

$$\frac{\partial}{\partial m}\mathbb{E}_{Y \sim P_x}[\log \mathcal{N}(Y; m, \Sigma)] = \Sigma^{-1}(\mathbb{E}_{Y \sim P_x}[Y] - m),$$

which is zero at the maximum $m^*(x) = \mathbb{E}_{Y \sim P_x}[Y]$. To find the optimal $\Sigma$, let $S_x = \mathbb{E}[(Y - m^*)(Y - m^*)^\top]$. We can rewrite the expectation term as

$$\mathbb{E}[\log \mathcal{N}(Y; m^*, \Sigma)] = \frac{1}{2}\log \det \Sigma^{-1} - \frac{1}{2}\mathrm{tr}(\Sigma^{-1} S_x) + \mathrm{const},$$

which is strictly concave in $\Sigma^{-1} \succ 0$. Differentiating with respect to $\Sigma^{-1}$ we obtain

$$\frac{\partial}{\partial \Sigma^{-1}}\mathbb{E}_{Y \sim P_x}[\log \mathcal{N}(Y; m^*, \Sigma)] = \frac{1}{2}\Sigma - \frac{1}{2}S_x.$$

This quantity is zero at the maximum $\Sigma^* = \mathbb{E}_{Y \sim P_x}[(Y - m^*)(Y - m^*)^\top]$, and uniqueness follows from strict concavity. Thus, the optimal covariance is given by $\widehat{\Gamma} := \mathbb{E}_{Y \sim P_x}[(Y - m^*)(Y - m^*)^\top]$, the centred covariance of the neighbours of $x$. $\qquad \square$

This theorem justifies using the mean-centred local covariance of the diffusion kernel's neighbourhood as the statistically optimal covariance for a local Gaussian fit.

**Rescaling the carré du champ matrix** If the bandwidth of the kernel is set appropriately, the carré du champ matrix $\widehat{\Gamma}(x^{(i)})$ will capture the local shape of the density near a training point $x^{(i)}$. However, depending on the bandwidth, samples from $\mathcal{N}(x^{(i)}, \widehat{\Gamma}(x^{(i)}))$ may be far from $x^{(i)}$. To quantify this, we may diagonalise $\widehat{\Gamma} = Q^\top \mathrm{diag}(\lambda_1, ..., \lambda_d)Q$, where $Q$ is the orthonormal matrix of principal components of $\widehat{\Gamma}$, then $\widehat{\Gamma}$ represents Gaussian noise with variance $\lambda_i$ in the direction of the component $Q_i$. Then, if samples are drawn randomly from $\mathcal{N}(x, \widehat{\Gamma})$, their expected squared distance from $x$ is

$$\mathbb{E}\left(\|X - x\|^2\right) = \mathrm{tr}(\widehat{\Gamma}).$$

This means that excessive noise in directions normal to the manifold can reduce the fidelity of the geometric regularisation. Thus, to control the maximum amount of noise added to the data by $\widehat{\Gamma}$ in any direction, we need to rescale $\widehat{\Gamma}$ such that the largest eigenvalue $\lambda_1$ is of the right scale.

As a heuristic, we would like the noise added by $\widehat{\Gamma}(x^{(i)})$ to be small enough that it is contained in the gaps between the training point $x^{(i)}$ and its neighbours. This way, the training signal is still dominated by the *location* of the training data, with the carré du champ adding a small, unintrusive amount of *directional* training on top of that. If $\pi(x^{(i)})$ is the nearest neighbour of $x^{(i)}$, then we choose to ensure that all the eigenvalues of $\widehat{\Gamma}(x^{(i)})$ are no larger than $\|x^{(i)} - \pi(x^{(i)})\|^2/9$. This means that, in any given direction, the noise added has standard deviation at most $\|x^{(i)} - \pi(x^{(i)})\|/3$, so at least 99.7% of it will be closer to $x^{(i)}$ than its nearest neighbour $\pi(x^{(i)})$.

This local heuristic is effective except in the case that $x^{(i)}$ is very isolated, in which case $\|x^{(i)} - \pi(x^{(i)})\|$ is very large. To avoid adding too much excess noise to these points, we will also cap the maximum noise level at $c_{\max}$, defined to be the $90^{th}$ percentile of all the local constraints $\|x^{(i)} - \pi(x^{(i)})\|^2/9$. This limits the scale of the noise at the 10% most outlying points in the data, which we can guarantee by ensuring that all the eigenvalues are below $c_{\max}^2$.

To meet both the local and global constraints, we rescale $\widehat{\Gamma}(x^{(i)})$ by

$$c_i = \frac{1}{\lambda_1^i} \min\left(\frac{\|x^{(i)} - \pi(x^{(i)})\|^2}{9}, c_{\max}^2\right), \tag{33}$$

where $\lambda_1^i$ is the largest eigenvalue of $\widehat{\Gamma}(x^{(i)})$. The largest eigenvalue of the rescaled matrix will then be the smaller of $\|x^{(i)} - \pi(x^{(i)})\|^2/4$ and $c_{\max}^2$.

The above method describes a heuristic for setting the size of the carré du champ at each point, but, in particular cases, we may attain better performance by further globally rescaling $\widehat{\Gamma}(x_i)$. We therefore use the carré du champ $\gamma c_i \widehat{\Gamma}(x^{(i)})$, where $c_i$ is from (33) and $\gamma$ is a tuneable hyperparameter that defaults to 1.0.

## F  COMPUTATIONAL COMPLEXITY AND RUNTIMES

The conditional paths in CDC-FM require $\widehat{\Gamma}(x_1)^{1/2}$, for which we have to compute and diagonalise $\widehat{\Gamma}(x_1)$. Since we truncate the diffusion kernel to have only $k$ non-zero entries per point (the neighbours of the point), the carré du champ at each point has rank at most $k$, so we can avoid working with full $d \times d$ covariance matrices. Instead, we project the neighbour differences into their $k$-dimensional span and compute $\widehat{\Gamma}(x_1)$ in this reduced basis. The resulting $k \times k$ matrices capture the full spectrum while being far cheaper to store and diagonalise. The dominant costs are therefore $O(k^2d)$ for the projection step and $O(k^3)$ for the eigen-decomposition, rather than $O(d^3)$. In practice, $k$ is small even when the dimension $d$ is high, so the computation effectively takes $\mathcal{O}(N(\log(N) + d))$ time and $\mathcal{O}(Nd)$ space. The complexities of the different steps are given in Table A1. We also report the empirical preprocessing time for the CIFAR-10 experiments in Table A2.

Table A1: **Compute and memory complexity.** Here, $N$ is the number of training points, $k$ is the number of neighbours, and $d$ is the ambient dimension.

| Step | Compute | Memory |
|---|---|---|
| k-NN graph construction | $\mathcal{O}(N\log(N))$ | $\mathcal{O}(Nk + Nkd)$ |
| Diffusion kernel | $\mathcal{O}(Nk)$ | $\mathcal{O}(Nk)$ |
| Form covariance terms in $\mathbb{R}^d$ | $\mathcal{O}(Nkd)$ | $\mathcal{O}(Nkd)$ |
| Project covariance terms to $\mathbb{R}^k$ | $\mathcal{O}(Nk^2d)$ | $\mathcal{O}(Nkd + Nk^2)$ |
| Diagonalise $\widehat{\Gamma}(x_i)$ in $\mathbb{R}^k$ | $\mathcal{O}(Nk^3)$ | $\mathcal{O}(Nk^2)$ |
| **Overall complexity** | $\mathcal{O}\big(N(\log(N) + k^2d + k^3)\big)$ | $\mathcal{O}(Nkd + Nk^2)$ |

Table A2: **Preprocessing empirical runtimes.** Runtimes are reported in seconds for CIFAR-10 subsets of varying sizes and were run on NVIDIA A10s.

| Dataset size | KNN graph (s) | $\widehat{\Gamma}$ eigendecomposition (s) | Total (s) |
|---|---|---|---|
| 1000 | 0.45 | 19.27 | 19.72 |
| 2000 | 0.55 | 38.03 | 38.58 |
| 3000 | 1.19 | 56.14 | 57.33 |
| 4000 | 1.10 | 73.56 | 74.66 |
| 5000 | 1.89 | 92.09 | 93.98 |

**Computational Resources**  All experiments were feasible and primarily performed on NVIDIA A10s (24 GB). Some experiments were performed on NVIDIA H100s (80GB). *Drosophila* experiments were performed on NVIDIA A100s (40GB).

## G  EXPERIMENT DETAILS

**Numerical solver.**  Unless stated otherwise, for inference and likelihood computations, we use the adaptive step size solver `dopri5` with `atol=rtol=1e-5` using the `torchdiffeq` library (Chen et al., 2019).

**Circle, two circles, LiDAR, and torus experiments.** These experiments use the same basic MLP architecture from Lipman et al. (2024) with 4 hidden layers of dimension 512 and Swish activations. We used the Adam optimiser with a learning rate of $10^{-3}$. We summarise the CDC-specific hyperparameters in Table A3. For the memorisation metric, we used a nearest-neighbour ratio cutoff of 0.2.

**Single cell experiment.** For the single cell experiments, we followed the setup in Kapusniak et al. (2024) and used a 3-layer MLP with width 1024 and SeLU activation to learn the vector field. We used the AdamW optimiser with a learning rate of $10^{-3}$ and a weight decay of $10^{-5}$. For inference, we used the Euler solver for 100 steps. We summarise the CDC-specific hyperparameters in Table A4. We found that adding isotropic noise with variance $\sigma_{\min}$ in addition to the anisotropic CDC noise helped stabilise training.

***Drosophila* experiment.** For the *Drosophila* experiments, we learn the vector field with a transformer architecture previously used for flow-matching-based human motion synthesis (Hu et al., 2023). To accommodate the simpler motion of *Drosophila* compared to human motion, we downscaled the network and used 4 transformer blocks with feedforward and key/value dimensions $d_{\mathrm{ff}} = d_{\mathrm{kv}} = 256$ with 4 attention heads and GELU activation. The model was trained on 2k training points for $10^5$ epochs using the Adam optimiser with a constant learning rate of $10^{-4}$ and batch size of 512. We summarise the CDC-specific hyperparameters in Table A3. To evaluate NLL, percentage memorised, and sample quality, we used a test set of 25k points as well as 25k generated samples. For the memorisation metric, we used a nearest-neighbour ratio cutoff of 0.6 (Fig. A2). As in DeAngelis et al. (2019), we fitted UMAP with 30 nearest neighbours on standardised data points, first subtracting the per-limb-coordinate mean across time steps from each trajectory and then scaling each of the 372 dimensions across all data points to zero mean and unit variance. UMAP was fitted on the larger test set, and the resulting model was subsequently used to project the training points into the UMAP embedding space.

**CIFAR-10 experiment.** For the CIFAR-10 experiments, we followed the setup in Tong et al. (2024) with the exception that we did not use the standard flip transform for data augmentation to focus on the fundamental differences between FM and CDC-FM losses. We used a UNet (Dhariwal & Nichol, 2021) with 128 channels, depth of 2, channel multiples $[1, 2, 2, 4]$, 4 heads, 64-channel heads, attention resolution of 16, and 0.1 dropout. Following Tong et al. (2024), we used a constant learning rate $2 \times 10^{-4}$, gradient clipping with norm 1.0, and exponential moving average weights with decay 0.9999.

We summarise the CDC-specific hyperparameters in Table A5. To tune the $d_{cdc}$ and $\gamma$ hyperparameters, we computed the empirical covariances as described in Section 3.2, we used the closed-form $p_1$ to generate 10k samples and computed the FID on a validation set consisting of 10k images. For negative log likelihoods, we report bits per dimension as done in Lipman et al. (2024) using the regular uniform dequantization with $K = 1$. FID is computed using the inception v3 embeddings (Szegedy et al., 2016) and the cleanfid package (Parmar et al., 2022). For the memorisation metric, we used a nearest-neighbour ratio cutoff of 0.2.

**Celeba-HQ experiment.** For the Celeba-HQ, we followed the setup in Dao et al. (2023) with the exception that we did not use the standard flip transform for data augmentation to focus on the fundamental differences between FM and CDC-FM losses. We use a UNet (Dhariwal & Nichol, 2021). For NLL we report bits per dimension of a test set in latent space. For FID computation we use the inception v3 embeddings (Szegedy et al., 2016). The CDC-FM-specific parameters were $k = 256$, $k_{bw} = 8$, $d_{cdc} = 4$, $\gamma = 0.5$.

Table A3: **Model parameters for different experiments.**

| Parameter | Description | Experiments | | | | |
|---|---|---|---|---|---|---|
| | | **Circle** (Fig. 1) | **LiDAR** (Fig. 2) | **Two-circles** (Fig. 3) | *Drosophila* (Fig. 4) | $d$-**Torus** (Fig. 5) |
| k | Maximal nearest neighbours $w_\epsilon$ | 3 | 32 | 3 | 128 | 32 |
| $k_{bw}$ | Bandwidth of $w_\epsilon$ | 8 | 8 | 8 | 8 | 8 |
| $d_{cdc}$ | Rank of $\widehat{\Gamma}$ | 1 | 2 | 1 | 2, 4, 8, 16 | $d$ |
| $\gamma$ | Scaling of $\widehat{\Gamma}$ | 0.3 | 1.0 | 0.7 | 0.3 | 1.0 |

Table A4: **Model parameters for the single-cell experiment.**

| Parameter | Description | Single Cell Experiments | | | |
|---|---|---|---|---|---|
| | | **Cite** | **Cite + OT** | **Multi** | **Multi + OT** |
| k | Maximal nearest neighbours $w_\epsilon$ | 256 | 256 | 256 | 256 |
| $k_{bw}$ | Bandwidth of $w_\epsilon$ | 8 | 8 | 8 | 8 |
| $d_{cdc}$ | Rank / CDC dimension | 8 | 4 | 4 | 2 |
| $\gamma$ | Scaling of $\widehat{\Gamma}$ | 0.5 | 0.4 | 0.4 | 0.1 |
| $\sigma_{\min}$ | Isotropic noise | 0.4 | 0.3 | 0.2 | 0.4 |

Table A5: **Model parameters for the different CIFAR-10 experiments.**

| Parameter | Description | Dataset size | | | | | | | |
|---|---|---|---|---|---|---|---|---|---|
| | | **250** | **500** | **750** | **1000** | **2000** | **3000** | **4000** | **5000** |
| k | Maximal nearest neighbours $w_\epsilon$ | 256 | 256 | 256 | 256 | 256 | 256 | 256 | 256 |
| $k_{bw}$ | Bandwidth of $w_\epsilon$ | 8 | 8 | 8 | 8 | 8 | 8 | 8 | 8 |
| $d_{cdc}$ | Rank / CDC dimension | 8 | 16 | 16 | 8 | 8 | 8 | 16 | 8 |
| $\gamma$ | Scaling of $\widehat{\Gamma}$ | 2.0 | 0.9 | 0.8 | 1.0 | 1.0 | 0.7 | 0.5 | 0.7 |

# H    Supplementary tables

Table A6: **Comparison between FM and CDC-FM on LiDAR landscape after 4k epochs.**

| Dataset Size | Generalisation (NLL) ↓ | | Memorisation ↓ | | Numerical efficiency (NFE) ↓ | | Distance to manifold ↓ | |
| --- | --- | --- | --- | --- | --- | --- | --- | --- |
| | FM | CDC-FM | FM | CDC-FM | FM | CDC-FM | FM | CDC-FM |
| 40 | 2.26 | **1.92** | 5.1 | **3.2** | **74** | **74** | **105** | 122 |
| 80 | **1.51** | 1.52 | 2.1 | **1.8** | **68** | **68** | **116** | 122 |
| 120 | **1.25** | 1.28 | 1.9 | **1.8** | **68** | **68** | **99** | 106 |
| 160 | **1.20** | 1.22 | **1.5** | **1.5** | 62 | 68 | **101** | 106 |
| 200 | **1.10** | 1.12 | 1.8 | **1.6** | **74** | **74** | **76** | 80 |

Table A7: **Comparison between FM and CDC-FM on LiDAR landscape after 16k epochs.**

| Dataset Size | Generalisation (NLL) ↓ | | Memorisation ↓ | | Numerical efficiency (NFE) ↓ | | Distance to manifold ↓ | |
| --- | --- | --- | --- | --- | --- | --- | --- | --- |
| | FM | CDC-FM | FM | CDC-FM | FM | CDC-FM | FM | CDC-FM |
| 40 | 3.50 | **2.23** | 32.4 | **7.5** | 92 | **80** | **65** | 101 |
| 80 | 2.16 | **1.66** | 15.4 | **6.5** | 98 | **86** | **56** | 72 |
| 120 | 1.65 | **1.45** | 11.0 | **6.3** | **98** | 104 | **50** | 62 |
| 160 | 1.30 | **1.22** | 4.8 | **3.3** | 98 | **92** | **56** | 65 |
| 200 | 1.34 | **1.24** | 5.6 | **3.5** | **98** | **98** | **49** | 56 |

# I    Negative Log-Likelihood Computation

To evaluate the generalisation capability of our models, we compute the negative log-likelihood (NLL) on held-out test data. Recall that the flow matching model defines a vector field $u_t(x)$ that generates a probability path $p_t(x)$ transforming a source distribution $p_0$ (typically $\mathcal{N}(0, \mathbf{I})$) to the model distribution $p_1 \approx \nu$. The log-density of a test point $x$ under the model is computed using the instantaneous change of variables formula for Continuous Normalising Flows (Chen et al., 2019):

$$\log p_1(x) = \log p_0(\psi_1^{-1}(x)) - \int_0^1 \nabla \cdot u_t(\psi_t(\psi_1^{-1}(x))) \, dt, \tag{34}$$

where $\psi_t$ is the flow map generated by $u_t$, and the integral is taken along the characteristic path from $t = 0$ to $t = 1$.

In high dimensions, explicitly computing the divergence $\nabla \cdot u_t = \mathrm{tr}(\partial u_t / \partial x)$ is computationally expensive ($\mathcal{O}(d^2)$). To ensure numerical stability and scalability, we employ the Hutchinson trace estimator (Hutchinson, 1989), which approximates the divergence using random noise vectors $\epsilon \sim \mathcal{N}(0, \mathbf{I})$:

$$\nabla \cdot u_t(x) = \mathbb{E}_\epsilon \left[ \epsilon^\top \frac{\partial u_t(x)}{\partial x} \epsilon \right]. \tag{35}$$

This reduces the complexity to $\mathcal{O}(d)$ using vector-Jacobian products. In our experiments, we solve the combined ODE for the particle trajectory and the log-density change using the `torchdiffeq` library with an adaptive step-size solver (`dopri5`) and tolerances of $10^{-5}$.

## J    SUPPLEMENTARY FIGURES

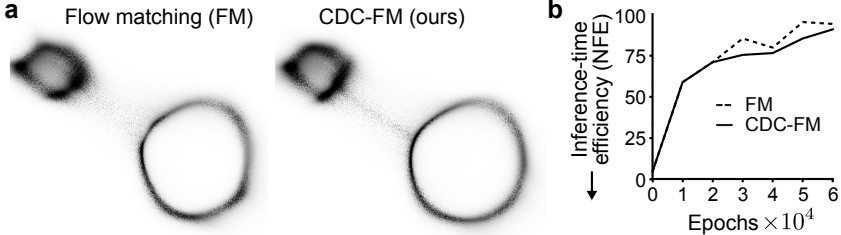

Figure A1: **Supplementary figures for the two-circles experiment.** **a** Samples from FM and CDC-FM, trained on the two-circles dataset at an epoch early in the training (10k), when the quality (distance to manifold) is still low, but generalisation is high. **b** Inference-time numerical efficiency of FM and CDC-FM.

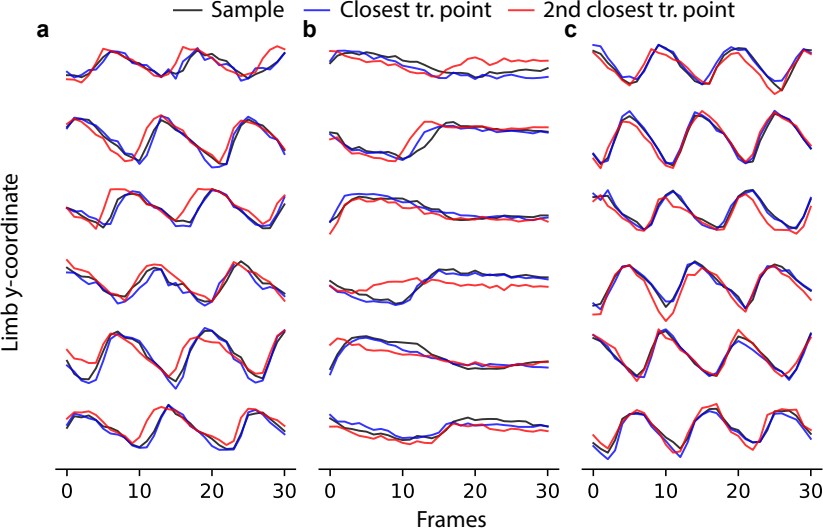

Figure A2: **Memorised samples in the *Drosophila* motion capture dataset.** Three sample time series showing the longitudinal motion for the six limbs for **a** $M = 0.6$, **b** $M = 0.6$ and **c** $M = 0.58$.

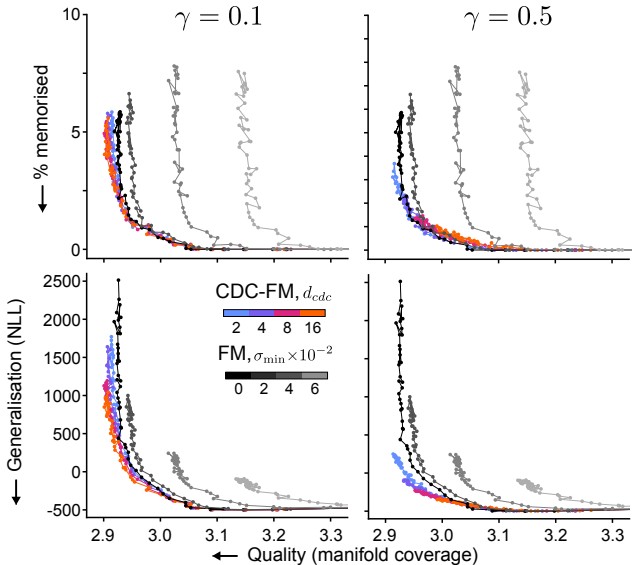

Figure A3: **Quality-generalisation and quality-memorisation for the *Drosophila* motion capture data.** Same as Fig. 4b,c, but with CDC rescaling parameter $\gamma = 0.1, 0.5$.

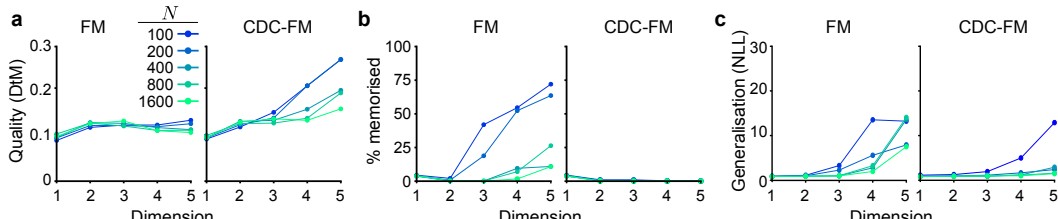

Figure A4: **Synthetic experiment on toroidal manifold with additive Gaussian noise.** Effect of data dimension on **a** generalisation, **b** memorisation and **c** sample quality. Noise level: $\sigma = 0.02$.

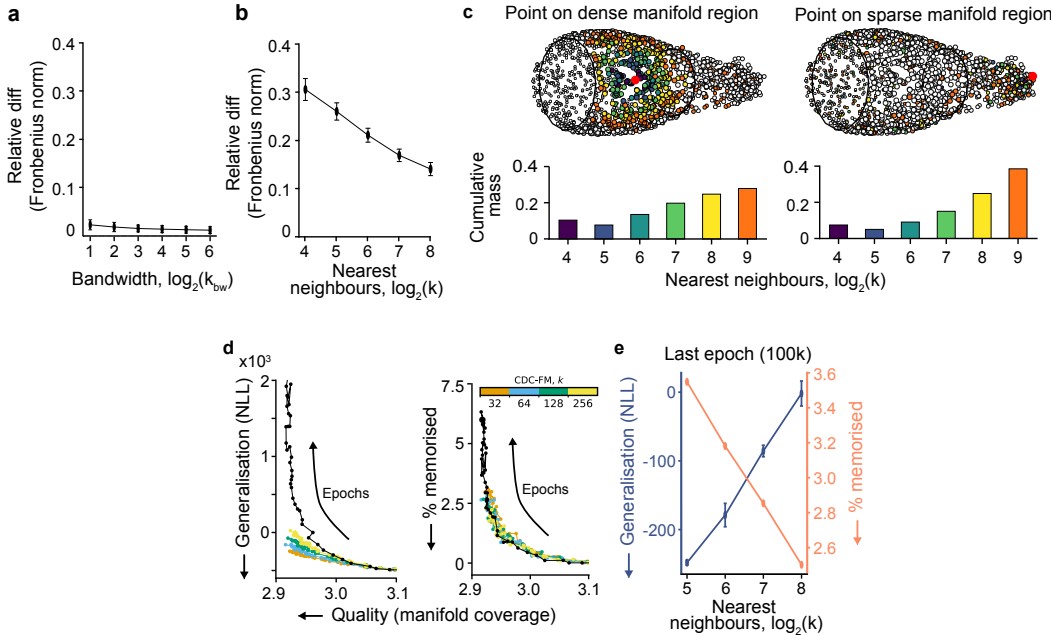

Figure A5: **Ablation studies corresponding to Fig. 4. a** Effect of doubling the bandwidth $k_{bw}$ on the CDC field at a given value ($k = 128$). Minimal effect observed. **b** Effect of doubling the number of nearest neighbours $k$ on the CDC field at a given value ($k_{bw} = 8$). The CDC field is less sensitive to changes in $k$ for higher $k$ values. **c** Neighbourhood of a point in a dense (left) and sparse (right) manifold region. Sparse regions can lead to shortcut connections that do not respect the manifold geodesic distance. The kernel density estimate assigns large mass to these points, biasing the CDC estimate. **d** Influence of the choice of $k$ on the overall generalisation and memorisation performance. CDC-FM provides gains over FM for all choices of $k$. **e** Generalisation and memorisation at the final epoch (100k, mean and std over 3 seeds). The overall performance gain (although improved compared to FM for all choices of $k$) is in a tradeoff between generalisation and memorisation. Small $k$ leads to better tangent space estimates and thus better generalisation, but yields less strong regularisation. Large $k$ leads to worse tangent space estimates, so generalisation drops, but overall reduces memorisation.

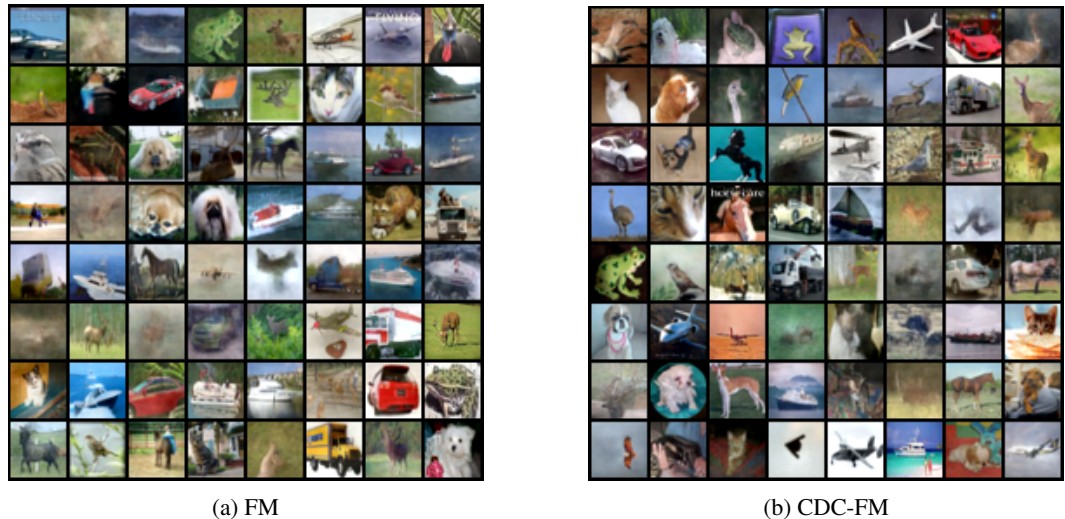

(a) FM
(b) CDC-FM

Figure A6: **Visual comparison of generated images from FM and CDC-FM models.** Models were trained 40k epochs on 5k images from CIFAR-10. Both models have similar visual quality.

