# OpenReview forum: "Carré du champ flow matching: better quality-generalisation tradeoff in generative models"
_ICLR.cc/2026/Conference — ICLR 2026 Poster_

### Official Review · Reviewer_M4z6 · 2025-10-31

**Soundness:** 3
**Presentation:** 3
**Contribution:** 3
**Rating:** 6
**Confidence:** 4

**Summary:**

This paper introduces **Carré du Champ Flow Matching (CDC-FM)**, a *geometry-aware generalization of Flow Matching (FM)* designed to improve the tradeoff between **sample quality** and **generalisation** in deep generative models.

Standard FM often suffers from *memorisation*, in which the model reproduces training examples rather than sampling broadly from the underlying manifold. CDC-FM addresses this by introducing a **spatially varying, anisotropic diffusion term** aligned with the data manifold’s local geometry.

The method provides both a **theoretical framework** linking geometric regularisation and generative transport, and a **scalable algorithm** (O(N log N)) applicable to large datasets.

**Strengths:**

1. **Conceptual Novelty and Theoretical Depth**
   - Reformulating flow matching as a geometry-regularised stochastic process via the *Carré du Champ* operator is mathematically elegant and well justified.
   - The theoretical analysis clearly connects the anisotropic diffusion term to Dirichlet energy minimisation and optimal transport interpolants.

2. **Empirical Breadth**
   - Demonstrations span multiple domains (geometric point clouds, biological data, motion capture, image synthesis), highlighting robustness.
   - Consistent improvements in *generalisation* metrics (NLL, Earth Mover’s Distance, memorisation ratio) show the practical benefit of geometry-aware noise.

3. **Quality–Generalisation Tradeoff Analysis**
   - The paper systematically studies how FM overfits under data sparsity and heterogeneous sampling and shows CDC-FM mitigates memorisation even without early stopping.
   - Figures such as Fig. 3–6 effectively illustrate the dynamics of quality, memorisation, and generalisation over training epochs.

**Weaknesses:**

1. **Limited Theoretical Generality**
   - The approach depends on accurate tangent-space estimation via diffusion maps, which scales poorly with high-dimensional data and may break for non-manifold or mixed-domain datasets.
   - The analysis assumes a strong form of the manifold hypothesis; the method’s behaviour in non-geometric or noisy data regimes is not deeply studied.

2. **Empirical Limitations**
   - Although CIFAR-10 results are included, they remain moderate; improvements appear strongest in low-data or geometric domains rather than large-scale image generation.
   - Some evaluations (e.g., motion-capture experiments) are largely qualitative; more statistical metrics (e.g., variance, confidence intervals) would improve credibility.

3. **Ablation and Hyperparameter Sensitivity**
   - The role of parameters such as the scaling factor \(\gamma\) and rank \(d_{cdc}\) is discussed but not systematically ablated.
   - The influence of kernel bandwidth and nearest-neighbour size on \(\hat{\Gamma}\) estimation could be quantitatively explored.

**Questions:**

1. How sensitive is CDC-FM to the choice of kernel bandwidth and neighbour count in the diffusion map estimation of \(\hat{\Gamma}\)?
2. Could the authors explore combining CDC regularisation with learned adaptive manifolds or latent diffusion embeddings?
3. In higher-dimensional settings (e.g., CIFAR-10), can CDC-FM’s benefits be retained through hierarchical or local manifold approximations?
4. How does CDC-FM interact with implicit architectural regularisation (e.g., transformers vs UNet backbones)?

---

> ### Author Response · Authors · 2025-11-25
> **Rebuttal (1/2)**
>
> We thank the reviewer for their thorough and insightful review, and for the positive assessment of our conceptual contribution, theoretical depth, and broad empirical study. We also appreciate the constructive suggestions regarding ablations and hyperparameter sensitivity, which we address below.
>
> ________
>
> > **W1:** The approach depends on accurate tangent-space estimation via diffusion maps, which scales poorly with high-dimensional data and may break for non-manifold or mixed-domain datasets.
>
> We agree that estimating tangent spaces in high dimensions requires an exponentially increasing number of data points, and we expect the benefit of CDC-FM to be greater in lower intrinsic dimensions. However, **we also highlight that scaling to high-dimensional data is non-trivial - from the perspective of this article, it amplifies the tradeoff between quality and memorisation.** Thus, while CDC-FM performance drops as the dimension increases (Fig. 5a), FM memorisation rapidly increases with dimension. In practice, we find that compared to FM, incorporating geometric information via our method is beneficial, even in high-dimensional examples (Table 1, Fig. 4b-d, Fig. 6).
>
> We clarified this in the paper. See the next point for a discussion of ‘non-manifold and mixed domains’.
>
> > **W2** The analysis assumes a strong form of the manifold hypothesis; the method’s behaviour in non-geometric or noisy data regimes is not deeply studied.
>
> We agree that our central modelling assumption is that the data distribution has locally low-dimensional structure. However, let us clarify that we use a weak form of the manifold assumption, based upon Markov diffusion processes to estimate the local tangent spaces, rather than, e.g., local PCA. These stochastic manifolds are:
> 1. **robust to noise**, as demonstrated by the synthetic experiment in Fig. A4.
> 2. **have freedom to vary in intrinsic dimension across the data**, the d_{CDC} hyperparameter being only an upper bound, although we do see that tuning $d_{CDC}$ improves performance (Fig. 4b).
> 3. **can be estimated locally based on local heat kernels**, which optimally adapt to the data sparsity (see Theorem 2 in Appendix E and Theorem 1 in Berry & Harlim (2015))
>
> Further, **our validation experiments** on real-world datasets from diverse domains, including computer vision (LiDAR, Fig. 2, Tables A6-A7), cell biology (gene expression trajectories, Table 1), image generation (Fig. 6) and motion capture (Fig. 4), **show that the manifold assumption does not compromise the broad practical applicability of our model.**
>
> > **W3:** Although CIFAR-10 results are included, they remain moderate; improvements appear strongest in low-data or geometric domains rather than large-scale image generation.
>
> We generally agree with this comment. We included this dataset because a significant fraction of generative modelling focuses on image domains. Yet, our main motive was to address scientific domains where datasets are more geometric and comparatively smaller. These domains are currently underserved by generative modelling, but the number of application areas is vast. We only highlight a few (single-cell, computer vision and motion capture), but there are a significant number of other domains, which are out of the scope of this article, including dynamical systems, molecule/protein design, etc., where memorisation is a more imminent issue, which will benefit from geometric frameworks.
>
> > **W4:** Some evaluations (e.g., motion-capture experiments) are largely qualitative; more statistical metrics (e.g., variance, confidence intervals) would improve credibility.
>
> Thank you for the feedback. Statistical errors in our model can arise from several sources
> 1. network weight initialisation
> 2. sampling during training
> 3. using a finite number of samples to estimate probability distributions used in generalisation, memorisation and quality metrics
>
> **To quantify the propagation of error in 1-2, we now retrained the networks from multiple seeds**. We reported confidence intervals in Fig. 4d and in the new Figs. S5. To avoid cluttering the results, we report averages over seeds in Fig. 4b,c, while noting that the confidence intervals over these results are small and strengthen our results.
>
> To estimate the errors arising from 3, we now also performed bootstrapped estimates of the quality/memorisation and generalisation metrics. This process resulted in no appreciable error bars, and, therefore, we did not report these values in additional figures.
>
> > **W5** The role of parameters such as the scaling factor (\gamma) and rank (d_{cdc}) is discussed but not systematically ablated.
>
> The scaling factor $\gamma$ and rank $d_{cdc}$ are both ablated in the motion-capture experiments, see Fig. 4b-d and Fig. A3

---

> > ### Author Response · Authors · 2025-11-25
> > **Rebuttal (2/2)**
> >
> > > **W6/Q1:** The influence of kernel bandwidth and nearest-neighbour size on (\hat{\Gamma}) estimation could be quantitatively explored. How sensitive is CDC-FM to the choice of kernel bandwidth and neighbour count in the diffusion map estimation of (\hat{\Gamma})?
> >
> > Thank you for this remark. In response, we have now ablated both the kernel bandwidth (k_{bw}) and number of nearest neighbours (k) parameters for the motion capture experiment (Fig. 4).
> >
> > We found that the k_{bw} had a negligible influence on the tangent space estimation (new Fig. A5a).
> >
> > The dependence of tangent space estimation on k relies on good practices from manifold learning. On the one hand, choosing k too small can result in few points being averaged over and thus suboptimal tangent space estimation, resulting in the increase of memorisation. On the other hand, choosing the k too high can result in ‘shortcut’ connections across the manifold, which do not coincide with geodesics (new Fig. A5c).
> >
> > However, **despite the possibility of tuning the k hyperparameter, we find that the performance of CDC-FM consistently exceeded that of FM across orders of magnitude changes in k (new Fig. A5e).**
> >
> > However, overall, we see that while tuning k can achieve higher performance gains, the latter remain higher than FM for all reasonable choices.
> >
> > > **Q2:** Could the authors explore combining CDC regularisation with learned adaptive manifolds or latent diffusion embeddings?
> >
> > This is an excellent suggestion, and we have added an experiment comparing FM and CDC-FM in the latent space of the pretrained Stable Diffusion VAE applied to a subset of 1000 high-resolution (256x256) images from the Celeba-HQ dataset. We find that CDC-FM improves both FID and NLL after 2k epochs when the model performance stabilises. We have added this experiment to the main text in Section 4.3 and Table 2, which we repeat here for completeness.
> >
> > | Epoch | FM | CDCFM |
> > | -------- | -------- | -------- |
> > | 1000     |   15.6   | 12.72     |
> > | 2000     | 13.51     | 13.42     |
> > | 3000     | 13.56     | 10.55     |
> > | 4000     | 13.82     | 11.70     |
> > | 5000     | 13.10     | 10.85     |
> >
> > Table 1: FID per epoch for FM and CDCFM in latent space. Celeba-HQ subset size 1000.
> >
> >
> > | Epoch | FM | CDCFM |
> > | -------- | -------- | -------- |
> > | 1000     |   6.80   | 7.18     |
> > | 2000     | 6.78     | 6.83     |
> > | 3000     | 6.80     | 6.53     |
> > | 4000     | 6.69     | 6.53     |
> > | 5000     | 6.68     | 6.48     |
> >
> > Table 2: NLL per epoch for FM and CDCFM in latent space. Celeba-HQ subset size 1000.
> >
> >
> > > **Q3:** In higher-dimensional settings (e.g., CIFAR-10), can CDC-FM’s benefits be retained through hierarchical or local manifold approximations?
> >
> > For our tests on CIFAR-10, we work in pixel space, but use a U-NET backbone that is inherently hierarchical due to the CNN pooling. This has a beneficial effect for both FM and CDC-FM.
> >
> > > **Q4:** How does CDC-FM interact with implicit architectural regularisation (e.g., transformers vs UNet backbones)?
> >
> > This is an interesting research question. Memorisation and generalisation depend on three factors: a) loss function, b) data and c) architecture. In this work, we specifically focused on a-b. Our central result is that modifying the loss function (through the CDC-FM conditional path, Eq. 8) allows us to reduce memorisation for a given generative quality. We also show that data sparsity patterns can significantly impact memorisation (Fig. 3, Fig. 4e-g), which our choice of conditional paths can reduce.
> >
> > Regarding architecture, our approach was to use, to our knowledge, the most expressive architectures for the given application domains. We reasoned that this would provide the strongest baseline inductive bias, which our methodology should improve upon.
> >
> > While we hypothesise that the choice of architecture is also an important determinant of memorisation, studying this would entail systematically controlling for model complexity and data geometry. Simply testing different architectures would not lead to sufficiently strong conclusions, and thus, this question is out of scope for this work.
> >
> > However, we added a comment on this in the Discussion.
> >
> >
> > ___________
> > We thank the reviewer for their time and insightful review, and for the positive assessment of our conceptual and empirical contributions. In light of the revisions and additional experiments, we would be grateful if they would consider re-evaluating the paper.

---

### Official Review · Reviewer_gnWi · 2025-10-31

**Soundness:** 4
**Presentation:** 3
**Contribution:** 3
**Rating:** 8
**Confidence:** 4

**Summary:**

The paper introduces Carré du Champ Flow Matching (CDC-FM), a generalization of the Flow Matching method designed to enhance generalization while maintaining generation quality. The approach incorporates a geometry-aware noise term as a regularizer. Its effectiveness is demonstrated through extensive experiments across diverse datasets, validating the model’s properties and performance.

**Strengths:**

- The paper is well-structured and clearly written, with a rich set of figures that effectively illustrate the concepts and accurately present the results.
- The work presents a novel, interesting, and mathematically sound solution to an open problem in flow matching, demonstrating strong practical performance.
- The model is extensively evaluated to properly experimentally show the benefits provided by the method, as well as to test and show the limitations, which are the ones usually affecting geometric models.

**Weaknesses:**

- The main weakness of the paper, also acknowledged by the authors, lies in its reliance on the manifold hypothesis. The experiments show that as the dimensionality of the data manifold increases, CDC-FM struggles to maintain the same sample quality as FM when trained with the same number of data points. I consider this a significant limitation, though the paper remains of high quality overall.

- Another area for improvement is the writing. Although the paper is detailed and mathematically precise, the motivation for adopting this specific approach to address the generalization–memorization problem in FM is not clearly articulated in the introduction. Moreover, while the method is rigorously developed in the introduction and Section 3, providing an intuitive explanation before delving into the technical discussion (e.g., before Equation (1)) would make the paper more accessible. I will elaborate on these points in the Questions section.

**Questions:**

**Experiments**
- In Tables A6 and A7, I noticed that the DtM for FM is consistently lower than for CDC-FM, which is quite interesting. Do you think this is just due to memorization?
- The experiment "Early stopping for spatially heterogeneous data" is very interesting.

**Paper structure and writing**
- In the introduction, the motivation for using geometric regularization as a means to improve the quality–generalization trade-off is not clearly stated. While it is clear that it introduces a more geometry-aware noise, it remains unclear how this would contribute to better generalization.
- Providing an intuitive explanation of the method - similar to the excellent one you give in the Conclusion - before presenting the technical details in the introduction would make the paper easier to follow.
- Making your loss function explicit could also improve clarity.
- The paragraph preceding Section 3.2, which introduces the connection to the Dirichlet energy and carré du champ, is quite dense but conceptually important. I believe this explanation should be expanded and made less synthetic.
- The explanation of the experiment on animal motion capture data is also rather dense and not very smooth to read. In this case, the solution might not be to add more text, but rather to move some details to the Appendix. Of course, this is just a suggestion to improve readability.

---

> ### Author Response · Authors · 2025-11-25
> **Rebuttal (1/2)**
>
> We thank the reviewer for their thoughtful and detailed review, and for the positive evaluation of our manuscript. We appreciate the recognition of our mathematical formulation and broad experimental study, as well as the constructive suggestions on motivation and clarity. We address the comments below.
>  _______
>
> > **W1:** The main weakness of the paper, also acknowledged by the authors, lies in its reliance on the manifold hypothesis.
>
> While we agree that our framework assumes the existence of a low-dimensional subspace, it considers a general class of manifolds defined locally as Markov processes, through their infinitesimal generators, the carré du champ. **Our stochastic manifolds provide robustness to noise, variable manifold dimensionality data sparsity patterns**. Please see ‘Response to all reviewers’ above for details. In sum, the manifold hypothesis for us is more of an opportunity than a weakness. We clarified this in the main text and the Discussion.
>
> In response to this concern, we have now performed a new experiment where we generate high-resolution (high-dimensional) images in latent space in the CelebA-HQ dataset. We find that CDC-FM consistently improves the quality and generalisation of FM. **This shows that dimensionality can be tamed by widely adopted latent space approaches**.
>
> > **W2:** The experiments show that as the dimensionality of the data manifold increases, CDC-FM struggles to maintain the same sample quality as FM when trained with the same number of data points. I consider this a significant limitation, though the paper remains of high quality overall.
>
> We must clarify that **our message was not only that scaling CDC-FM to high D is challenging (Fig. 4a, but also that but also that flow matching is more prone to memorisation (Fig. 5b)**. Specifically, Fig. 5b shows that the higher quality achieved by FM in dimensions 3 and higher coincides with almost 100% of the training points being memorised. Memorised points achieve perfect quality, at the cost of a complete loss of generalisation. Our key message is that a generative model should obtain high quality without memorising (i.e. copying training points). Thus, overall, compared to FM, incorporating geometric information via this method is beneficial, even in high-dimensional examples (Table 1, Fig. 4b-d, Fig. 6).
>
> We have now clarified this point in the main text.
>
> > **W3:**  Another area for improvement is the writing.[...]  I will elaborate on these points in the Questions section.
>
> Thank you for the feedback. We have now rewritten the Introduction, providing more intuition. Please see below for further details.
>
> ### *Experiments*
> __________
> > **Q1:** In Tables A6 and A7, I noticed that the DtM for FM is consistently lower than for CDC-FM, which is quite interesting. Do you think this is just due to memorization?
>
> Indeed, we also find it interesting that FM achieves higher quality than CDC-FM in the LiDAR example, both early in the training (4k epochs, Table A6) and late in the training (16k epochs, Table A7). Late in the training, memorisation is significantly higher in FM vs. CDC-FM (Table A7, 32.4% vs. 7.5% for 40 data points and 5.6% vs. 3.5% for 200 data points). Early in the training, data points are, in general, not memorised for FM and CDC-FM. However, quality, while higher for FM, is generally low. Thus, while FM’s bias towards quality versus memorisation is interesting, early training epochs are not very representative of the fine structure in the data.
>
> > **Q2:** The experiment "Early stopping for spatially heterogeneous data" is very interesting.
>
> Thank you for the positive comment. Indeed, we found the ‘two-circles’ dataset a very useful toy model to illustrate local memorisation.

---

> > ### Author Response · Authors · 2025-11-25
> > **Rebuttal (2/2)**
> >
> > ### **Writing**
> >
> > > **Q3:** In the introduction, the motivation for using geometric regularization as a means to improve the quality–generalization trade-off is not clearly stated. While it is clear that it introduces a more geometry-aware noise, it remains unclear how this would contribute to better generalization.
> >
> > Thank you for bringing this to our attention. Previous works (Ross et al. 2025, Achilli et al. 2024 and Ventura et al. 2025) have found that memorisation coincides with the intrinsic dimensionality of the data manifold suddenly dropping to zero. Effectively, the generative model learns an empirical distribution composed of a collection of discretely supported probability masses, rather than a continuous density. Our geometric intuition is to enforce the model to learn the tangent spaces of dimension strictly greater than zero (i.e., the memorised case). We achieve this by estimating the local covariances (the carré du champ) and using them to define a new conditional probability path that preserves the data manifold.
> >
> > We have clarified this in the second paragraph of the Introduction.
> >
> > > **Q4:** Providing an intuitive explanation of the method - similar to the excellent one you give in the Conclusion - before presenting the technical details in the introduction would make the paper easier to follow.
> >
> > We now provide more intuition in the Introduction, as explained in the previous reply.
> >
> > > **Q5:** Making your loss function explicit could also improve clarity.
> >
> > Our loss function is the same as the conditional flow matching loss (Eq. 6), but with the conditional probability path replaced by Eq. 8.
> >
> > We clarified this in the text after equation (9).
> >
> > > **Q6:** The paragraph preceding Section 3.2, which introduces the connection to the Dirichlet energy and carré du champ, is quite dense but conceptually important. I believe this explanation should be expanded and made less synthetic.
> >
> > Thank you for the feedback. We have now rewritten Section 3 to provide a clearer explanation of the connection between our augmented flow paths and the Dirichlet energy.
> >
> > > **Q7:** The explanation of the experiment on animal motion capture data is also rather dense and not very smooth to read. In this case, the solution might not be to add more text, but rather to move some details to the Appendix. Of course, this is just a suggestion to improve readability.
> >
> > We have now rewritten this section to improve clarity, and thank the reviewer for the suggestion.
> >
> > _______
> >
> > We thank the reviewer once again for their time, careful reading, and especially for the thoughtful suggestions on improving the motivation and clarity of the manuscript.

---

### Official Review · Reviewer_bx3v · 2025-11-01

**Soundness:** 3
**Presentation:** 3
**Contribution:** 3
**Rating:** 6
**Confidence:** 4

**Summary:**

The paper addresses a key challenge in deep generative modeling via flow-matching methods: achieving very high sample quality often comes at the cost of over-fitting or memorization of the training data, rather than genuine generalization to the underlying data manifold. The authors propose a novel method termed Carré du champ flow matching (CDC-FM) that augments the standard flow‐matching (FM) framework with a geometry‐aware noise regularization.

**Strengths:**

The paper is well written. Fig. 1 provides an overview of the proposed method and clearly shows its difference with classic flow matching.

They introduce the novel idea of aligning the noise covariance in the conditional paths of FM with the local geometry of the data manifold. It leverages diffusion geometry / local covariance estimation to regularise generative flows. The authors also show the connection with anisotropic diffusion, providing nice theoretical justification.

The authors show how one can estimate the local covariance (noise field) from data in a scalable way (via k-NN/diffusion kernel methods) and integrate it into the FM pipeline. The method can be plugged into existing flow matching implementations.

The experiments span a nice variety of domains: synthetic engineered manifolds, point clouds, genomics, motion capture, and images. o	According to the results, CDC‐FM achieves similar or better sample quality while reducing memorisation and improving generalisation.

**Weaknesses:**

The authors point out the scalability issue due to the use of manifold hypothesis. When the underlying manifold dimension grows, the samples required to estimate the tangent space grow exponentially.

The practical benefit seems strongest in specialized regimes (maybe low-dimensional manifold data, low data) rather than the large‐scale image domain that most practitioners focus on; the implementation adds complexity and hyper-parameters.

**Questions:**

1. Is there a practical guidance on when to use Carré du champ FM vs regular FM?

2. In practice, people use latent diffusion models (or flow-based models) for high-dimensional data.Can Carré du champ FM work in the latent space? Is it possible to show some results on data like CelebA-HQ?

---

> ### Author Response · Authors · 2025-11-25
> **Rebuttal**
>
> We thank the reviewer for their thoughtful review and positive assessment, particularly regarding the clarity of our formulation, the theoretical justification, and the breadth of experiments. We address the reviewer’s concerns below.
>
> ______
> > **W1:** The authors point out the scalability issue due to the use of manifold hypothesis. When the underlying manifold dimension grows, the samples required to estimate the tangent space grow exponentially.
>
> We agree that estimating tangent spaces in high dimensions is challenging, and we expect the benefit of CDC-FM to be greater in lower intrinsic dimensions. However, we must clarify that **our message was not only that scaling CDC-FM to high D is challenging (Fig. 4a, but also that FM achieves scaling by increased memorisation (Fig. 5b)**. Memorised points achieve perfect quality, at the cost of a complete loss of generalisation. Thus, overall, compared to FM, incorporating geometric information via our method is beneficial, even in high-dimensional examples (Table 1, Fig. 4b-d, Fig. 6).
>
> At the request of the reviewer, **we have now performed another experiment in which we generate high-resolution (high-dimensional) images in a low-dimensional latent space**. See point 4 below for details.
>
> > **W2** The practical benefit seems strongest in specialized regimes (maybe low-dimensional manifold data, low data) rather than the large‐scale image domain that most practitioners focus on; the implementation adds complexity and hyper-parameters.
>
> We generally agree with this remark. A large part of our motivation was to improve performance in scientific domains that are underserved by current methods. We feel that the focus of practitioners on the large-scale image domain is partly because this is where the existing methods work the best, and we wanted to expand the scope of the methods.
>
>
> > **Q1**  Is there a practical guidance on when to use Carré du champ FM vs regular FM?
>
> We thank the reviewer for the insightful question. Our experiments suggest that CDC-FM is most beneficial in data-scarce or geometry-sensitive settings, where FM is more prone to overfitting. While delineating the cases where CDC-FM is guaranteed to improve performance is challenging, the best advice would be to try and test. This is because for $\gamma=0$, the two models should coincide. Further, using CDC-FM does not add a significant computational overhead (Appendix F), and the generative model remains scalable. So, slowly increasing regularisation would generally be a practical strategy.
>
> > **Q2** In practice, people use latent diffusion models (or flow-based models) for high-dimensional data. Can Carré du champ FM work in the latent space? Is it possible to show some results on data like CelebA-HQ?
>
> This is also an excellent suggestion, and we have added an example using CDC-FM in the latent space for CelebA-HQ. We use the Stable Diffusion pretrained VAE to generate latent embeddings for a subset of 1000 high-resolution CelebA-HQ images (256x256). We then train both FM and CDC-FM on the embeddings and decode the output back to pixel space. We evaluate the models using FID and NLL at different training points. We find that CDC-FM outperforms FM on metrics after 2k epochs once both performances stabilise. This corroborates our previous results and shows that CDC-FM works in the latent space. For completeness, we add the results in the Tables below.
>
>
> | Epoch | FM | CDCFM |
> | -------- | -------- | -------- |
> | 1000     |   15.6   | 12.72     |
> | 2000     | 13.51     | 13.42     |
> | 3000     | 13.56     | 10.55     |
> | 4000     | 13.82     | 11.70     |
> | 5000     | 13.10     | 10.85     |
>
> Table 1: FID per epoch for FM and CDCFM in latent space. Celeba-HQ subset size 1000.
>
>
> | Epoch | FM | CDCFM |
> | -------- | -------- | -------- |
> | 1000     |   6.80   | 7.18     |
> | 2000     | 6.78     | 6.83     |
> | 3000     | 6.80     | 6.53     |
> | 4000     | 6.69     | 6.53     |
> | 5000     | 6.68     | 6.48     |
>
> Table 2: NLL per epoch for FM and CDCFM in latent space. Celeba-HQ subset size 1000.
>
> &nbsp;
> ________
>
> We thank the reviewer for their time, effort, and thoughtful review. In response to the reviewer’s comments, we added the suggested CelebA-HQ latent space experiment, and would be grateful if they could consider re-evaluating the paper in light of these revisions.

---

### Official Review · Reviewer_7Zem · 2025-11-01

**Soundness:** 3
**Presentation:** 3
**Contribution:** 3
**Rating:** 6
**Confidence:** 3

**Summary:**

The paper proposes CDC-FM, a geometry-aware variant of flow matching that replaces the standard isotropic conditional path with an anisotropic, spatially varying Gaussian whose covariance estimates local tangent structure via diffusion-geometry tools. The resulting conditional path is the displacement (OT) interpolant between standard Gaussian and the spatially varying Gaussian. The authors show is equivalent to adding a space-dependent diffusion (Fokker–Planck) term. Across geometric datasets (circles, LiDAR surfaces, single-cell trajectories, animal motion), CDC-FM aims to mitigate memorisation while preserving or improving sample quality and test NLL, especially in data-scarce and heterogeneous regimes.

**Strengths:**

1. Clear replacement of the FM conditional path with an OT-consistent Gaussian path. The equivalence to adding anisotropic diffusion (Fokker–Planck form) is neat and well-derived.

2. It works with MLP/UNet/Transformer and diverse domains, including both low-dim synthetic and higher-dim biological/motion data and scaling discussion.

3. The paper uses nearest-neighbour ratio to quantify memorisation; separates quality, generalisation, and the percentage memorised, showing heterogeneous behaviour over manifolds/regions.

**Weaknesses:**

1. The test NLL is model-dependent and sometimes hard to compute reliably for FM variants;

2. Key assertions (e.g., improved convergence/quality or robustness) are not tested across standard benchmarks, strong baselines, or multiple seeds with confidence intervals.

**Questions:**

1. How is NLL computed across models?

---

> ### Author Response · Authors · 2025-11-25
> **Rebuttal**
>
> We thank the reviewer for their thoughtful and constructive review. We appreciate the positive assessment of our work, and are grateful that the reviewer found the OT-based formulation clear, the Fokker–Planck interpretation neat, and the empirical coverage broad. We address each of the reviewer’s points below.
>
> ___________
> > **W1:** The test NLL is model-dependent and sometimes hard to compute reliably for FM variants
>
> We thank the reviewer for this question. The NLL of a model is a standard evaluation metric for generative models, and this is particularly true for flow models, including flow matching (see e.g. Lipman 2022). Unlike diffusion models, which require converting the SDE into an ODE and can introduce numerical instability, flow models provide direct access to the ODE, which gives access to the likelihood through the instantaneous change-of-variable formula. In particular, we follow Lipman 2022 and use the adaptive solver dopri5 with absolute and relative tolerance 10^-5, as well as the Hutchinson trace estimator for divergence estimation.
>
> We have added more details of our NLL computation in the **new Appendix I**.
>
> > **W2:** Key assertions (e.g., improved convergence/quality or robustness) are not tested across standard benchmarks, strong baselines, or multiple seeds with confidence intervals.
>
> Thank you for this feedback. We answer each point sequentially.
>
> *Baselines.*
>
> We agree with the reviewer that we focus exclusively on generalising FM, and not other flow models, by adding geometric noise to the flow paths. FM is broadly adopted by various communities, including generative modelling and robotics. Therefore, we reasoned that **FM is of sufficiently general importance to adopt it as our baseline**. Focusing on FM enabled us to delineate the mathematical connections and performance advantages in different toy models and diverse real-world benchmarks. We agree that other flow models can have diverse regularising behaviours, and we certainly cannot rule out another flow being more favourable. However, exploring all these analytical and numerical connections across other flow models is beyond the scope of this work.
>
> We have now sharpened the Discussion to better define the scope of our work and highlight the future need to extend to other flow models.
>
> *Benchmarks.*
>
> In response to this critique, **we have now performed an additional benchmark on a high-dimensional (~200k D), but small (1k) subset of the CelebA-HQ dataset**. Specifically, we used a pretrained VAE to map this dataset into a lower (~4k D) latent space, where we trained FM and CDC-FM. We found consistently better generalisation and quality (new Table 2) than FM. This demonstrates that our approach can be readily combined with latent space diffusion to tame high-dimensional datasets.
>
> *Multiple seeds with confidence intervals.*
>
> **We have now repeated the motion capture data experiments with multiple seeds and also performed error-propagation experiments from finitely sampled distributions (Fig. 4 and new Fig. S5).**
>
> These results further underline the statistical significance of our results.
>
>
> >  **Q1:** How is NLL computed across models?
>
> We use the method described in W1 and the new Appendix I for all models.
>
> _________
> We thank the reviewer for their time and effort in reviewing our paper. We have incorporated the requested clarifications and added new experiments, which we believe strengthen the work. We would be grateful if the reviewer could consider re-evaluating the paper in light of these revisions.

---

### Author Response · Authors · 2025-11-25
**General response**

# General response

We thank all reviewers for their positive evaluation of our manuscript and helpful suggestions. We especially appreciate the comments on the mathematical rigour, broad relevance and empirical breadth of our work.

Here we summarise the main criticisms, and our responses. We have also updated the PDF to reflect all changes.

__________

> ### **Concern 1:** The method is limited to data domains covered by the classical ‘manifold hypothesis’.

We generally agree with the reviewers that our central modelling assumption is that the data distribution has locally low-dimensional structure. **We formalise this assumption by approximating the data locally by Markov diffusion processes**, the carré du champ (CDC).

We emphasise that we use a weak form of the manifold assumption, based upon Markov diffusion processes, to estimate the local tangent spaces. These stochastic manifolds are

1. **robust to noise**, as demonstrated by the synthetic experiment in Fig. A4.
2. **have freedom to vary in intrinsic dimension across the manifold**, the $d_{CDC}$ hyperparameter being an upper bound on the manifold dimension, although we do see that tuning $d_{CDC}$ improves performance (Fig. 4b).
3. **can be estimated locally based on local heat kernels**, which optimally adapt to the data sparsity (see Theorem 2 in Appendix E and Theorem 1 in Berry & Harlim (2015))

Further, **our validation experiments** on real-world datasets from diverse domains, including computer vision (LiDAR, Fig. 2, Tables A6-A7), cell biology (gene expression trajectories, Table 1), image generation (Fig. 6, Table 2) and motion capture (Fig. 4), **show that the manifold assumption does not compromise the broad practical applicability of our model**.

We also envision that our model will directly impact other domains of central importance to generative modelling, including dynamical systems, molecule and protein design, where the manifold hypothesis holds and where memorisation is a pressing issue. However, this is beyond the scope of this work.

&nbsp;

> ### **Concern 2:** Applicability to high-dimensional data

The reviewers raised that the performance of our model drops as the ambient dimension increases. We agree with this, and we expect the benefit of CDC-FM to be greater in lower-dimensional manifolds.

However, we must clarify that **our message was not only that scaling CDC-FM to high dimensions is challenging (Fig. 4a), but also that but also that flow matching is more prone to memorisation (Fig. 5b)**.


This observation does not change our experimental results that, compared to FM, incorporating geometric information via CDC-FM is beneficial, even in high-dimensional examples (Table 1, Fig. 4b-d, Fig. 6).

To further demonstrate the scaling with dimensionality, we now conducted an additional experiment, where we mapped the approx. 200k-dimensional CelebA-HQ dataset to an approx 4k-dimensional latent space where we trained CDC-FM. We find that we obtain significant quality and generalisation improvements compared to FM (**new Table 2**). This demonstrates that **CDC-FM can be readily adapted to latent diffusion frameworks, which is a typical approach to address high dimensionality**.

&nbsp;
> ### **Concern 3:** Sensitivity analyses and uncertainty estimates

In response to this critique, **we now provide additional sensitivity analysis and error propagation experiments**. Specifically, in the motion capture experiment (Fig. 4), we performed the following experiments:
1. Retrained our model from different seeds and provided confidence intervals around our model predictions (**new Fig. 4b-d**).
2. Performed additional ablation experiments on the parameters k, $\gamma$ and k_{bw} (**new Fig. S5**).

These experiments conclude that our model maintains an advantage over FM across different choices of parameters.

&nbsp;

> ### **Concern 4:** Clarity of writing.

We now provide more intuitive motivation in the Introduction before the mathematical exposition.

___________
We thank all reviewers for their time, effort, and constructive comments. We provide detailed responses to individual reviewer comments below.

---

### Author Response · Authors · 2025-12-04

Given the unfortunate circumstances and the additional work they place on the AC, we provide a short summary of the reviews and rebuttal in the hope that it is useful.

All four reviewers gave positive scores (6,6,6,8) and highlighted the paper’s **conceptual novelty, mathematical rigour, and broad empirical evaluation across diverse domains.**

The main criticisms were focused on three areas

1. Applicability in high-dimensional settings and latent space (bx3v, gnWi, M4z6)
2. Statistical rigour: multiple seeds, confidence intervals, hyperparameter sensitivity (7Zem, M4z6)
3. Conceptual clarification of definitions and better intuitive motivation in the introduction (gnWi, 7Zem)

In the rebuttal we addressed all of these points with:
- new CelebA-HQ latent-space experiments showing consistent FID/NLL improvements (new Table 2);
- multi-seed runs, confidence intervals, and systematic ablations on parameters (k, γ, d_CDC) (new Figure A5, updated Figure 4);
- rewritten Introduction and Section 3 and extended methodology section (new Appendix I).


Because the discussion ended prematurely, reviewers did not have the opportunity to respond to the rebuttal. We think that the **new experiments and statistical evidence** we provide demonstrate the broad and immediate applicability of our work on the generative modelling community.

**We thank the AC again for their time, care, and additional effort under these circumstances.**

---

### Meta-Review · Area_Chair_pofn · 2026-01-07

**Summary:**

This paper proposes Carré du champ Flow Matching (CDC-FM), a generalization of standard Flow Matching (FM) that incorporates geometry-aware noise regularization. By replacing isotropic noise with anisotropic Gaussian noise estimated via the diffusion geometry (carré du champ operator) of the data manifold, the method aims to mitigate memorization while maintaining sample quality.
The reviewers were unanimously positive about the paper’s technical novelty, theoretical contributions, and the breadth of empirical evaluation across diverse scientific domains (genomics, motion capture, point clouds).
The primary concerns initially raised focused on the method's applicability to high-dimensional data (where the manifold hypothesis might struggle or tangent space estimation becomes expensive), the lack of standard latent-space benchmarks (e.g., CelebA-HQ), and requests for more statistical analysis (error bars/different seeds).
The paper is a strong contribution to generative modeling. It offers a principled solution to the memorization problem in FM, and a mathematical framework for studying data geometry and generalization.

**Reviewer Concerns:**

Overall, the authors provided a comprehensive rebuttal that effectively addressed most of the concerns.

**Addressed concerns**
- High-Dimensional/Latent Space Applicability (Reviewers bx3v, gnWi, M4z6): This was the most significant critique. Reviewers questioned if CDC-FM could scale beyond low-dimensional manifolds. The authors responded with a new experiment applying CDC-FM in the latent space of a pretrained VAE on CelebA-HQ (high-res images). The results (Tables 1 & 2 in rebuttal) demonstrated consistent improvements in FID and NLL over standard FM. This successfully demonstrated that the method is compatible with latent diffusion pipelines, and can be practically applied to high-dimensional images via latent-diffusion framework.
- Statistical significance (Reviewers 7Zem, M4z6): Concerns regarding single-seed runs and lack of confidence intervals were addressed by new multi-seed experiments on motion capture data and error-propagation analysis (Fig. A5)
- Ablation Studies (Reviewer M4z6): The reviewer asked for ablations on $\gamma$ and $d_{cdc}$. The authors pointed out that some were present but provided expanded ablations on kernel bandwidth and nearest-neighbors $k$ in the rebuttal, showing the method is robust to hyperparameter choices.
- Clarity(gnWi): The authors rewrote the introduction and Section 3 to provide better intuition and make the paper easier to follow.

**Outstanding concerns**
Reviewer 7Zem commented about the lack of strong baselines, while the authors argue that vanilla FM is a strong competitor, it still lacks comparison with other variants of FM or generative models that explicitly consider data geometry.

**Reviewer Scores:**

Based on the rebuttal and the already positive initial reviews, I predict that reviewers would maintain their positive consensus, with reviewers requesting results from latent diffusion raising their scores, given the new results.

- Reviewer 7Zem (Original: 6): Predicted: 6 (Marginally Above). The reviewer’s major concern about FM NLL computation is valid, and the mentioned weakness about including "strong baselines" were not addressed fully.
- Reviewer bx3v (Original: 6): Predicted: 8 (Accept). This reviewer explicitly asked: "Can Carré du champ FM work in the latent space?" The authors provided data showing it outperforms FM on CalebA-HQ. This should have boosted their confidence in the paper's empirical significnace.
- Reviewer gnWi (Original: 8): Predicted: 8 (Accept). This reviewer was already a champion and mostly asking clarification questions and suggesting better writing for accessibility. The rebuttal answers the questions and improves writing to some extent.
- Reviewer M4z6 (Original: 6): Predicted: 6 (Marginally Above). While the rebuttal addressed their points, the review flagged with LLM usage was somewhat generic. The additional ablations provided likely sustain the score, but might not drive a strong increase.

---

### Decision · Program_Chairs · 2026-01-26

Accept (Poster)